# Munc13 supports fusogenicity of non-docked vesicles at synapses with disrupted active zones

Chao Tan[1], Giovanni de Nola[1], Claire Qiao[1], Cordelia Imig[2,3], Richard T Born[1], Nils Brose[3], Pascal S Kaeser[1]*

[1]Department of Neurobiology, Harvard Medical School, Boston, United States; [2]Department of Neuroscience, University of Copenhagen, Copenhagen, Denmark; [3]Department of Molecular Neurobiology, Max Planck Institute for Multidisciplinary Sciences, Goettingen, Germany

**Abstract** Active zones consist of protein scaffolds that are tightly attached to the presynaptic plasma membrane. They dock and prime synaptic vesicles, couple them to voltage-gated $Ca^{2+}$ channels, and direct neurotransmitter release toward postsynaptic receptor domains. Simultaneous RIM + ELKS ablation disrupts these scaffolds, abolishes vesicle docking, and removes active zone-targeted Munc13, but some vesicles remain releasable. To assess whether this enduring vesicular fusogenicity is mediated by non-active zone-anchored Munc13 or is Munc13-independent, we ablated Munc13-1 and Munc13-2 in addition to RIM + ELKS in mouse hippocampal neurons. The hextuple knockout synapses lacked docked vesicles, but other ultrastructural features were near-normal despite the strong genetic manipulation. Removing Munc13 in addition to RIM + ELKS impaired action potential-evoked vesicle fusion more strongly than RIM + ELKS knockout by further decreasing the releasable vesicle pool. Hence, Munc13 can support some fusogenicity without RIM and ELKS, and presynaptic recruitment of Munc13, even without active zone anchoring, suffices to generate some fusion-competent vesicles.

*For correspondence: kaeser@hms.harvard.edu

## Editor's evaluation

Tan and colleagues studied synaptic transmission, presynaptic protein levels, and synaptic ultra-structure in hippocampal cultures of mice lacking the key active-zone proteins RIM (1, 2), ELKS (1, 2), and Munc13 (1, 2). Compared to cultures lacking only RIM and ELKS, additional deletion of Munc13 results in a further decrease of synaptic Munc13-1 levels, a similar reduction of the number of docked synaptic vesicles, and a more pronounced decrease of the readily releasable vesicles. The results support the conclusion of the nonredundant role of Munc13 in synaptic vesicle priming. Overall, this study reinforces the notion that synapse formation is a remarkably resilient process that occurs even under strong perturbation of presynaptic function.

## Introduction

Neurotransmitter release is mediated by synaptic vesicle fusion at presynaptic active zones, and Munc13 proteins have a central role in this process (*Brunger et al., 2018*; *Dittman, 2019*; *Wojcik and Brose, 2007*). In addition to Munc13, active zone scaffolds contain RIM, ELKS, RIM-BP, Bassoon/ Piccolo, and Liprin-α. Together, they form a protein machine that controls the speed and precision of synaptic transmission by docking and priming of synaptic vesicles, by organizing the coupling of these vesicles to sites of $Ca^{2+}$ entry, and by targeting transmitter release toward postsynaptic receptor

domains (*Biederer et al., 2017*; *Südhof, 2012*). Given the molecular complexity of the active zone and the essential roles of several of its protein components, understanding its assembly and mode of action has remained both a key aim and a challenge in cellular neuroscience.

Mouse knockout studies established major roles for Munc13 in vesicle priming at central nervous system synapses (*Augustin et al., 1999*; *Varoqueaux et al., 2002*). This process renders vesicles fusion-competent and adds them to the pool of readily releasable vesicles (RRP) that can rapidly undergo exocytosis upon action potential arrival. The functional RRP can be probed experimentally by applying stimuli that deplete it, for example, superfusion with hypertonic sucrose solution (*Kaeser and Regehr, 2017*; *Rosenmund and Stevens, 1996*). When Munc13 is deleted, release competence of vesicles is abolished or strongly decreased across all tested synapses in multiple model organisms, including mouse, *Drosophila melanogaster,* and *Caenorhabditis elegans* (*Aravamudan et al., 1999*; *Augustin et al., 1999*; *Richmond et al., 1999*; *Varoqueaux et al., 2002*). Munc13 ablation also results in an almost complete loss of docked vesicles, as defined by plasma membrane attachment in electron micrographs (*Hammarlund et al., 2007*; *Imig et al., 2014*; *Siksou et al., 2009*). These findings led to the model that vesicle docking and priming are morphological and functional states that correspond to release competence, a notion that is supported by similar correlations upon ablation of SNARE proteins (*Chen et al., 2021*; *Hammarlund et al., 2007*; *Imig et al., 2014*; *Kaeser and Regehr, 2017*). Due to their core function in vesicle priming, Munc13s mediate the resupply of fusion-competent vesicles as they are spent during synaptic activity and thereby control short-term plasticity and recovery from synaptic depression (*Lipstein et al., 2013*; *Lipstein et al., 2021*; *Rosenmund et al., 2002*). Furthermore, Munc13 nano-assemblies may account for secretory hotspots and recruit the SNARE protein syntaxin-1 (*Reddy-Alla et al., 2017*; *Sakamoto et al., 2018*).

Several recent studies took the approach to ablate combinations of active zone protein families, allowing analyses of active zone assembly and function upon elimination of redundant components (*Acuna et al., 2016*; *Brockmann et al., 2020*; *Kushibiki et al., 2019*; *Oh et al., 2021*; *Tan et al., 2022*; *Wang et al., 2016*; *Zarebidaki et al., 2020*). We simultaneously deleted RIM1, RIM2, ELKS1 and ELKS2, which resulted in a loss of RIM, ELKS, and Munc13, and in strong decreases of Bassoon, Piccolo, and RIM-BP levels at active zones (*Tan et al., 2022*; *Wang et al., 2016*; *Wong et al., 2018*). This disruption of active zone assembly led to a near-complete loss of synaptic vesicle docking as studied by electron microscopy and to an ~85% decrease in single action potential-evoked exocytosis as assessed electrophysiologically. However, some transmitter release persisted: up to ~35% of release evoked by hypertonic sucrose and ~50% of spontaneous release events remained under the tested experimental conditions, increasing extracellular $Ca^{2+}$ to increase vesicular release probability p strongly enhanced single action potential-evoked release, and stimulus trains released vesicles surprisingly efficiently (*Wang et al., 2016*). Hence, some releasable vesicles persisted despite the loss of most docked vesicles. These findings supported alternative mechanistic models in which docking and priming are independent processes mediated by distinct molecular functions of Munc13 (*Kaeser and Regehr, 2017*). Further support for this model came from experiments with artificial retargeting of Munc13 to synaptic vesicles rather than to active zones, which increased vesicle fusogenicity but not docking in mutants that lack most RIM and ELKS sequences and docked vesicles (*Tan et al., 2022*). Together, these findings suggest that non-docked vesicles can contribute to the functional RRP after removal of RIM and ELKS. What remained enigmatic, particularly in view of the near-complete loss of Munc13 from the target membrane after RIM + ELKS ablation, was whether undocked vesicles engaged Munc13 for vesicle priming or used an alternative priming pathway.

In this study, we tested directly whether Munc13 is necessary to prime vesicles after the strong active zone disruption upon RIM + ELKS knockout. We generated mice to ablate Munc13-1 and Munc13-2 in addition to RIM and ELKS (that is RIM1, RIM2, ELKS1, and ELKS2) and compared phenotypes of hextuple knockout neurons with those from conditional RIM + ELKS knockouts only. We found that synapses formed despite this strong genetic manipulation and that overall their ultrastructure was largely normal except for a lack of docked vesicles. However, Munc13 ablation on top of RIM and ELKS knockout further impaired single action potential-evoked release and decreased the RRP at both excitatory and inhibitory synapses. Paired pulse ratios, used to monitor p, were not further affected by the additional removal of Munc13. Our data establish that Munc13 can functionally prime some vesicles in the absence of RIM and ELKS, indicate that Munc13 away from active zones is sufficient to confer vesicle fusogenicity, and support a growing body of evidence showing that synapse formation

is overall resilient to severe perturbations of synaptic protein content and of synaptic activity. We propose that Munc13 recruitment to presynapses is rate-limiting to generate fusion competence of synaptic vesicles.

## Results

### Some synaptic Munc13 remains after RIM + ELKS knockout

With the overall goal to determine whether Munc13 mediates addition of vesicles to the functional RRP after RIM + ELKS knockout, we first confirmed key effects on synaptic transmission and Munc13 active zone levels in cultured neurons after ablation of RIM + ELKS (*Figure 1*) that we had described before (*Tan et al., 2022*; *Wang et al., 2016*). We cultured primary hippocampal neurons of mice with floxed (fl) alleles for RIM1, RIM2, ELKS1 and ELKS2 (*Figure 1A*). At 5 days in vitro (DIV5), a time point that is before the detection of functional synapses in these cultures (*Held et al., 2020*; *Mozhayeva et al., 2002*), the neurons were transduced with Cre-expressing lentiviruses or control lentiviruses (that express an inactive mutant of cre) to generate cKO$^{R+E}$ neurons or control$^{R+E}$ neurons, respectively. We previously established that this induces strong defects in active zone assembly and neurotransmitter release, but the neurons form synapses at normal numbers, and postsynaptic receptor assemblies and functions are preserved (*Tan et al., 2022*; *Wang et al., 2016*). We first confirmed that excitatory and inhibitory synaptic transmission are strongly impaired in cKO$^{R+E}$ neurons (*Figure 1B–F*). Synaptic responses were induced by focal electrical stimulation, and whole-cell recordings served to monitor excitatory and inhibitory transmission via glutamate and GABA$_A$ receptors (GABA$_A$Rs), respectively. Action potential-evoked excitatory transmission was monitored via NMDA receptors (NMDARs) to prevent the strong reverberant activity that is induced by stimulation of these neuronal networks when AMPA receptors (AMPARs) are not blocked.

We next evaluated Munc13-1 positioning and levels at the active zone using stimulated emission depletion (STED) superresolution microscopy (*Figure 1G–K*). We stained for Synaptophysin to mark the synaptic vesicle cloud (imaged by confocal microscopy), PSD-95 to mark the postsynaptic density (PSD, imaged by STED), and Munc13-1 (imaged by STED). In these experiments, side-view synapses are defined as a synaptic vesicle cluster with an elongated PSD-95 structure aligned at one side of the vesicle cloud as described before (*Emperador-Melero et al., 2021a*; *Held et al., 2020*; *Nyitrai et al., 2020*; *Tan et al., 2022*; *Wong et al., 2018*). To evaluate Munc13-1 levels in the active zone area, we quantified PSD-95 and Munc13-1 fluorescence levels in 600 nm × 200 nm areas that were positioned perpendicular to the PSD through the center of the PSD-95 signal, and plotted their line profiles (*Figure 1H and I*) and peak levels (*Figure 1J and K*). Based on these analyses, Munc13-1 was largely lost from the active zone area of cKO$^{R+E}$ synapses.

For comparison, we analyzed Munc13-1 antibody signals at the active zone after ablation of Munc13. We used mice with floxed alleles for Munc13-1 and constitutive knockout alleles for Munc13-2 and Munc13-3 (*Figure 1—figure supplement 1A*; *Augustin et al., 2001*; *Banerjee et al., 2022*; *Varoqueaux et al., 2002*). In cultured neurons of these mice, Cre expression removes Munc13-1 (cKO$^M$) without the potential for compensation by Munc13-2 or -3. Control experiments were performed on the same cultures but with lentiviral expression of an inactive Cre (control$^M$). Munc13-1 was ablated efficiently in cKO$^M$ neurons (*Figure 1—figure supplement 1B–F*), with the leftover signal not distinguishable from background levels that are typically observed in this approach (*Held et al., 2020*; *Nyitrai et al., 2020*; *Wong et al., 2018*). When we compared Munc13-1 levels in cKO$^{R+E}$ and cKO$^M$ synapses, the remaining signal was somewhat higher in cKO$^{R+E}$ synapses (*Figure 1— figure supplement 1G*). These higher levels did not arise from a peak at the position of the active zone area (around –70 to –20 nm) (*Tan et al., 2022*; *Wong et al., 2018*). Instead, Munc13-1 levels appeared higher more broadly, and the ratio of Munc13-1 at cKO$^{R+E}$ vs. cKO$^M$ synapses shifted upward throughout the presynaptic bouton (*Figure 1—figure supplement 1G*). In both types of neurons, Synaptophysin signals remained unchanged (*Figure 1—figure supplement 1H and I*). As observed before in autaptic cultures (*Banerjee et al., 2022*), there was some release left at these Munc13 knockout synapses (*Figure 1—figure supplement 2*). This is likely due to the very small amount of remaining Munc13-1 when using this conditional allele (*Banerjee et al., 2022*).

Because some Munc13-1 could be detected in cKO$^{R+E}$ neurons by STED microscopy (*Figure 1G–K*, *Figure 1—figure supplement 1G*) and Western blotting (*Wang et al., 2016*), we compared synaptic

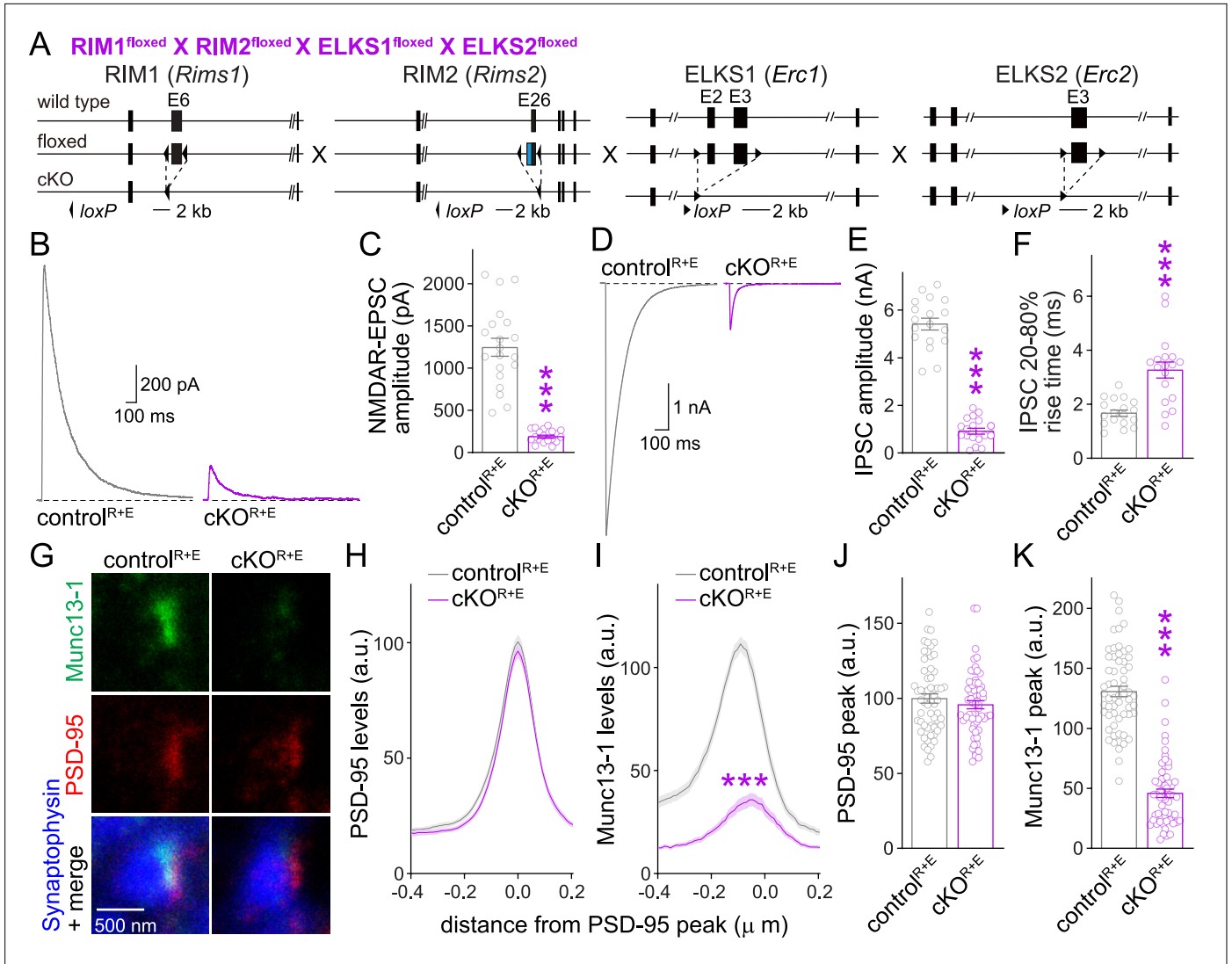

**Figure 1.** Action potential-evoked neurotransmitter release and Munc13 active zone levels after ablation of RIM + ELKS. (**A**) Strategy for deletion of RIM1, RIM2, ELKS1, and ELKS2 in cultured hippocampal neurons. Neurons of mice with floxed alleles for all four genes were infected with Cre-expressing lentiviruses (to generate cKO[R+E] neurons) or lentiviruses expressing a recombination-deficient version of Cre (to generate control[R+E] neurons) as described (*Tan et al., 2022*; *Wang et al., 2016*). (**B, C**) Sample traces (**B**) and quantification (**C**) of excitatory postsynaptic currents (EPSCs) evoked by focal electrical stimulation, control[R+E] 20 cells/3 cultures, cKO[R+E] 19/3. (**D–F**) Sample traces (**D**) and quantification of amplitudes (**E**) and 20–80% rise times (**F**) of inhibitory postsynaptic currents (IPSCs) evoked by focal electrical stimulation, 18/3 each. (**G–K**) Sample stimulated emission depletion (STED) microscopic images (**G**) and quantification (**H–K**) of side-view synapses in cultured hippocampal neurons stained for Munc13-1 (imaged in STED), PSD-95 (imaged in STED), and Synaptophysin (imaged in confocal). In (**H, I**), fluorescence levels at each position of line profiles (600 nm × 200 nm) positioned perpendicular to the center of the elongated PSD-95 structure and aligned to the PSD-95 peak are shown; in (**J, K**), peak values for each line profile are shown independent of peak position, 60 synapses/3 cultures each. Data are mean ± SEM; ***p<0.001 as determined by Welch's *t*-tests (**C, E, F**), two-way ANOVA followed by Bonferroni's multiple-comparisons post-hoc tests (**H, I**), or unpaired two-tailed Student's *t*-tests (**J, K**). For assessment of Munc13-1 levels after Munc13 knockout using STED microscopy, Synaptophysin levels in cKO[R+E] synapses, and comparison of Munc13-1 levels by STED microscopy in cKO[R+E] and cKO[M] synapses, see *Figure 1—figure supplement 1*; for assessment of synaptic transmission after Munc13 knockout, see *Figure 1—figure supplement 2*; for assessment of Munc13-1 expression by confocal microscopy and Western blotting in cKO[R+E] and cKO[M] neurons, see *Figure 1—figure supplement 3*. *Source data 1* contains numerical values of all means, errors, and p-values for this and all figures.

The online version of this article includes the following source data and figure supplement(s) for figure 1:

**Figure supplement 1.** Assessment of Munc13-1 levels with STED microscopy in Munc13 knockout synapses.

**Figure supplement 2.** Action potential-evoked neurotransmitter release after ablation of Munc13.

**Figure supplement 3.** Assessment of Munc13-1 levels with confocal microscopy and Western blot in RIM + ELKS or Munc13 knockout neurons.

**Figure supplement 3—source data 1.** Western blots for *Figure 1—figure supplement 3*.

Munc13-1 levels in cKO^{R+E} and cKO^M neurons (*Figure 1—figure supplement 3*). There were somewhat higher Munc13-1 signals in confocal microscopic images in cKO^{R+E} synapses compared to cKO^M synapses (*Figure 1—figure supplement 3A, B, D, and E*). Similarly, a slight Munc13-1 band was detected in Western blots of cKO^{R+E} neurons, but not of cKO^M neurons (*Figure 1—figure supplement 3C and F*). The remaining Munc13-1 signal in immunostainings detected in cKO^M neurons is likely mostly composed of antibody background in these experiments as quantifications were done without background subtraction and noise levels of ~25% are common (*Wang et al., 2016*). Altogether, our data indicate that some Munc13-1 might remain in nerve terminals of cKO^{R+E} synapses, but the remaining Munc13 is not efficiently concentrated in the active zone area apposed to the PSD, supporting previous data on RIM-mediated recruitment of Munc13 to the active zone (*Andrews-Zwilling et al., 2006*).

## Synapses form after deletion of Munc13 in addition to RIM and ELKS

With the overall goal to test whether Munc13-1 mediates the remaining release in cKO^{R+E} neurons, we crossed the conditional knockout mice for RIM1, RIM2, ELKS1, and ELKS2 to conditional Munc13-1 and constitutive Munc13-2 knockout mice (*Figure 2A*). Cultured hippocampal neurons from these mice were infected with Cre-expressing or control lentiviruses at DIV5 to generate cKO^{R+E+M} and control^{R+E+M} neurons, respectively, to remove Munc13 in addition to RIM + ELKS. We first used STED microscopy to analyze the localization and levels of RIM1 and Munc13-1 . RIM1 and Munc13-1 were effectively removed from active zones of cKO^{R+E+M} neurons (*Figure 2B–K*, *Figure 2—figure supplement 1A*). PSD-95 levels as judged by this method were unaffected in most quantifications (*Figure 2C, E, and J*), but a very small change was detected in *Figure 2H*. Because PSD-95 is used for synapse selection (see 'Materials and methods'), this isolated change likely reflects small differences in synapse selection in the two conditions in this specific experiment. The Munc13-1 signal that remained in STED experiments of cKO^{R+E} neurons (*Figure 1I and K*, *Figure 1—figure supplement 1G*) further decreased in cKO^{R+E+M} neurons (*Figure 2—figure supplement 1B*). In confocal images, there was a further decrease in synaptic Munc13-1 levels in cKO^{R+E+M} neurons compared to cKO^{R+E} neurons (*Figure 2—figure supplement 1C–E*), and Munc13-1 was below detection threshold in Western blotting (*Figure 2—figure supplement 1F*). Finally, Synaptophysin puncta number, size, and intensity, analyzed with a custom-written code to perform automatic two-dimensional segmentation for object detection (*Emperador-Melero et al., 2021a*; *Held et al., 2020*; *Liu et al., 2018*), were indistinguishable between control^{R+E+M} and cKO^{R+E+M} neurons (*Figure 2L–O*). This indicates that synapse formation is intact in the absence of the tested active zone proteins. Similar results were obtained when we analyzed synapses in neurons infected with Cre-expressing lentiviruses at DIV2 (*Figure 2—figure supplement 2*) instead of DIV5.

We next analyzed synapse ultrastructure using high-pressure freezing followed by transmission electron microscopy in the cultured neurons (*Figure 3*) with established procedures (*Tan et al., 2022*; *Wang et al., 2016*). In these analyses, docked synaptic vesicles are defined as vesicles for which the electron density of the vesicular membrane touches that of the presynaptic plasma membrane, and less electron-dense space cannot be detected between the two membranes. Simultaneous deletion of RIM, ELKS, and Munc13 abolished vesicle docking (*Figure 3A and B*) similar to RIM + ELKS ablation (*Tan et al., 2022*; *Wang et al., 2016*). While bouton size was unchanged (*Figure 3D*), there was a small decrease in vesicle numbers and a mild increase in the length of the PSD in cKO^{R+E+M} neurons (*Figure 3C and E*). This might be caused by homeostatic adaptations or by general roles of these proteins in synapse development, or be coincidental. Altogether, however, the morphological analyses, including STED, confocal and electron microscopy, establish that nerve terminals and synaptic appositions form and are overall ultrastructurally near-normal apart from a loss of docked vesicles despite the strong genetic manipulation with deletions of RIM1, RIM2, ELKS1, ELKS2, Munc13-1 and Munc13-2.

## Deletion of Munc13 in addition to RIM and ELKS further impairs synaptic vesicle release

Using whole-cell recordings, we then assessed synaptic transmission in cKO^{R+E+M} neurons and corresponding controls. We first measured spontaneous vesicle release by assessing miniature excitatory and inhibitory postsynaptic currents (mEPSCs and mIPSCs, respectively) in the presence of the sodium

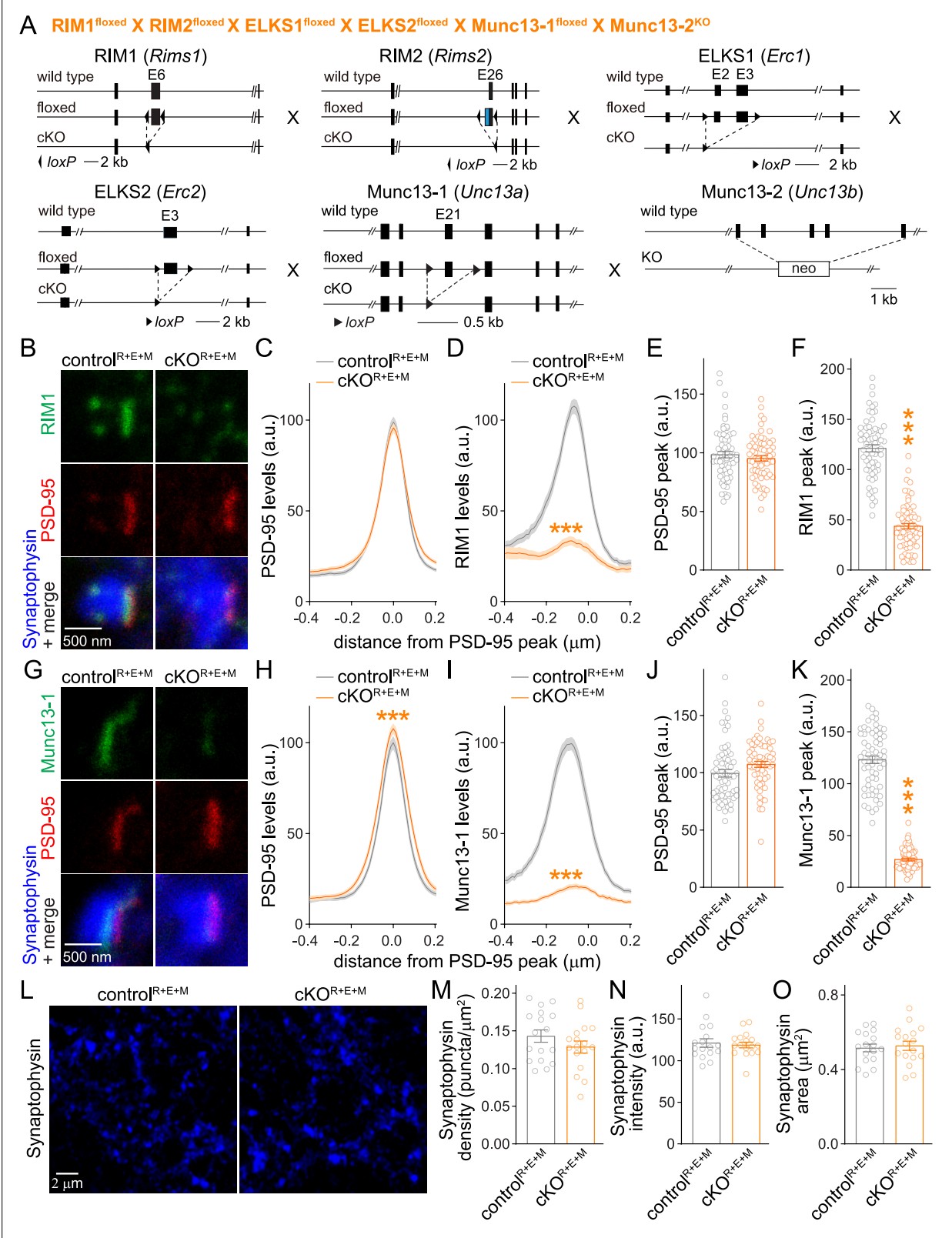

**Figure 2.** Simultaneous deletion of RIM, ELKS, and Munc13 does not disrupt synapse formation. (**A**) Strategy for simultaneous deletion of RIM1, RIM2, ELKS1, ELKS2, Munc13-1, and Munc13-2 in cultured hippocampal neurons (cKO[R+E+M]). Neurons were infected with Cre-expressing lentiviruses (to generate cKO[R+E+M] neurons) or viruses expressing a recombination-deficient version of Cre (to generate control[R+E+M] neurons). (**B–F**) Sample images (**B**) and quantification (**C–F**) of side-view synapses stained for RIM1 (STED), PSD-95 (STED), and Synaptophysin (confocal), control[R+E+M] 65 synapses/3

*Figure 2 continued on next page*

*Figure 2 continued*

cultures, cKO^(R+E+M) 66/3. (**G–K**) Same as (**B–F**), but for synapses stained for Munc13-1 instead of RIM1, 63/3 each. (**L–O**) Overview confocal images of anti-Synaptophysin staining (**L**) and quantification of Synaptophysin puncta density (**M**), intensity (**N**), and size (**O**); Synaptophysin objects were detected using automatic two-dimensional segmentation, confocal images of Synaptophysin staining are from the experiment shown in (**B–F**), 17 images/3 cultures each. Data are mean ± SEM; ***p<0.001 as determined by two-way ANOVA followed by Bonferroni's multiple-comparisons post-hoc tests (**C, D, H, I**), unpaired two-tailed Student's *t*-tests (**E, J, M–O**), or Welch's *t*-tests (**F, K**). For Synaptophysin levels in cKO^(R+E+M) synapses, comparison of Munc13-1 levels by STED microscopy in cKO^(R+E) and cKO^(R+E+M) neurons, and Munc13-1 expression in control^(R+E+M) and cKO^(R+E+M) neurons assessed by confocal microscopy and Western blotting, see *Figure 2—figure supplement 1*; for microscopic assessment of synapse formation after lentiviral infection at 2 days in vitro (DIV2) instead of DIV5, see *Figure 2—figure supplement 2*.

The online version of this article includes the following source data and figure supplement(s) for figure 2:

**Figure supplement 1.** Assessment of Munc13-1 levels in RIM + ELKS + Munc13 knockout neurons.

**Figure supplement 1—source data 1.** Western blots for *Figure 2—figure supplement 1*.

**Figure supplement 2.** Assessment of Synaptophysin fluorescence in RIM + ELKS + Munc13 knockout neurons after lentiviral infection at DIV2.

**Figure supplement 2—source data 1.** Western blots for *Figure 2—figure supplement 2*.

channel blocker tetrodotoxin. The frequencies of mEPSCs and mIPSCs were robustly decreased in cKO^(R+E+M) neurons compared to control^(R+E+M) neurons, while their amplitudes remained unchanged (*Figure 4A–F*). In addition, there was a small increase in mEPSC rise times, similar to cKO^(R+E) neurons (*Tan et al., 2022*), while mEPSC decay times and mIPSC kinetics were unchanged (*Figure 4—figure supplement 1*). Hence, vesicle release is impaired, but postsynaptic neurotransmitter detection is in essence intact in cKO^(R+E+M) neurons.

We then measured single action potential-evoked release at cKO^(R+E+M) synapses (*Figure 4G–K*). The cKO^(R+E+M) neurons had very strong reductions in evoked release, both at excitatory (*Figure 4G and H*)

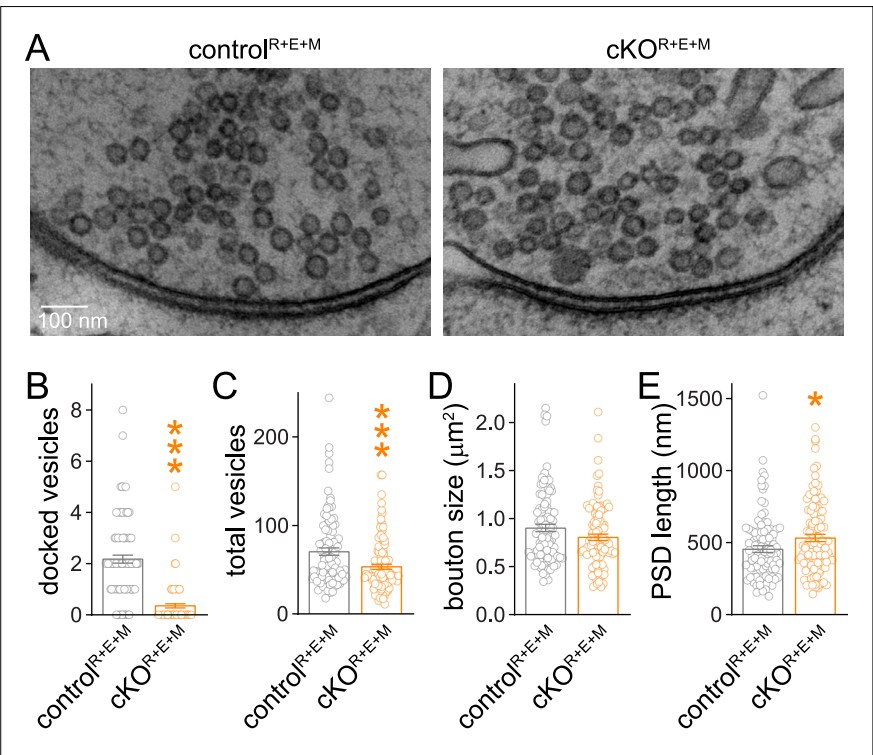

**Figure 3.** Synaptic ultrastructure after RIM + ELKS + Munc13 knockout. (**A–E**) Sample images (**A**) and analyses (**B–E**) of synaptic morphology of high-pressure frozen neurons analyzed by electron microscopy; docked vesicles (**B**), total vesicles (**C**), bouton size (**D**), and postsynaptic density (PSD) length (**E**) per synapse profile are shown. Docked vesicles were defined as vesicles for which the electron density of the vesicular membrane touched that of the target membrane such that the two membranes were not separated by less electron-dense space, control^(R+E+M) 99 synapses/2 cultures, cKO^(R+E+M) 100/2. Data are mean ± SEM; *p<0.05, ***p<0.001 as determined by Welch's *t*-tests (**B, C**) or unpaired two-tailed Student's *t*-test (**D, E**).

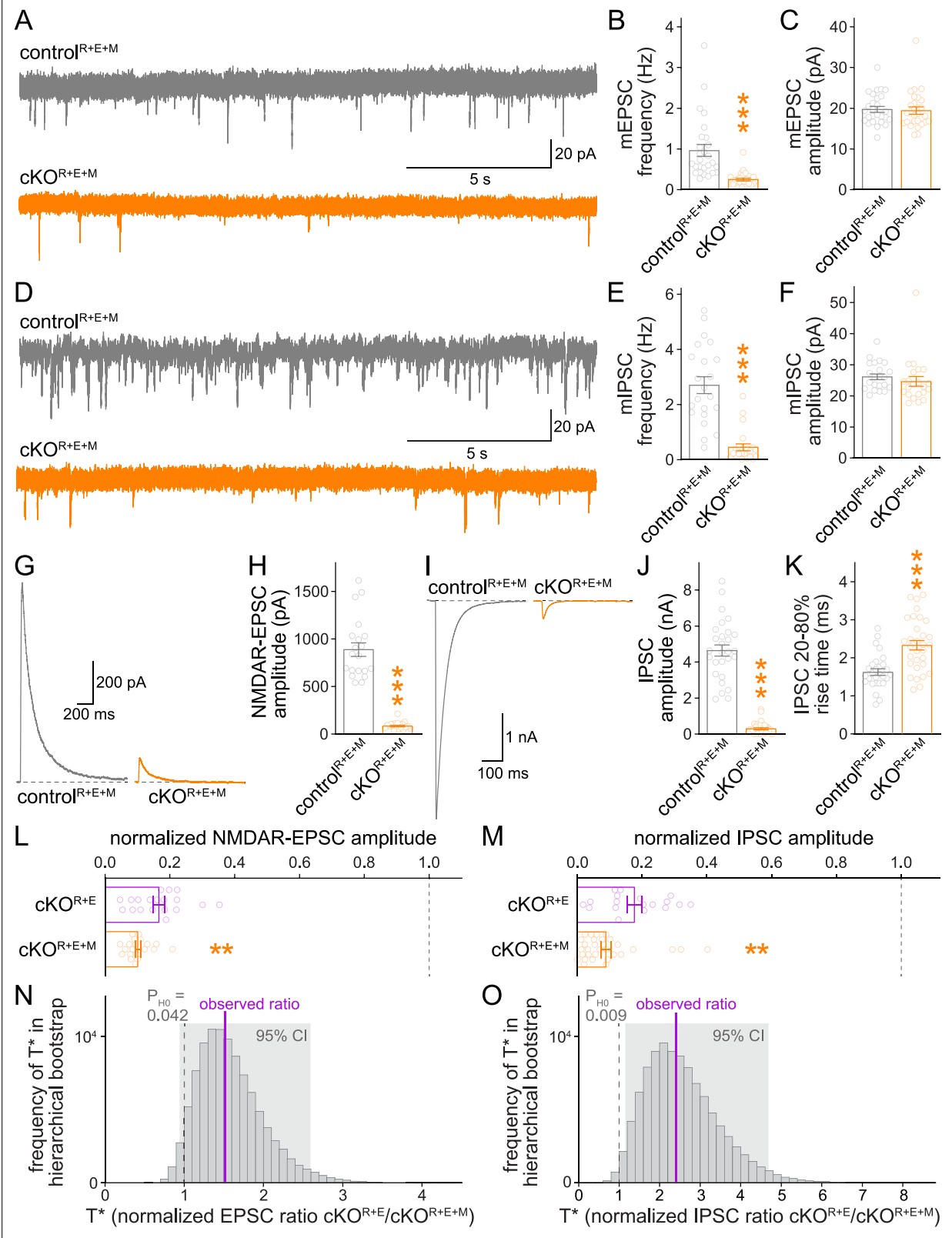

**Figure 4.** Neurotransmitter release is strongly impaired after RIM + ELKS + Munc13 ablation. (**A–C**) Sample traces (**A**) and quantification of miniature excitatory postsynaptic current (mEPSC) frequencies (**B**) and amplitudes (**C**) in control[R+E+M] and cKO[R+E+M] neurons, 27 cells/3 cultures each. (**D–F**) Sample traces (**D**) and quantification of miniature inhibitory postsynaptic current (mIPSC) frequencies (**E**) and amplitudes (**F**), 22/3 each. (**G, H**) Sample traces (**G**) and quantification (**H**) of EPSCs evoked by focal electrical stimulation, 20/3 each. (**I–K**) Sample traces (**I**) and quantification of amplitudes (**J**) and

*Figure 4 continued on next page*

*Figure 4 continued*

20–80% rise times (**K**) of IPSCs evoked by focal electrical stimulation, control[R+E+M] 28/3, cKO[R+E+M] 31/3. (**L**) Comparison of EPSCs normalized to their own controls for cKO[R+E] (absolute data from *Figure 1C*) and cKO[R+E+M] (from **H**) neurons, cKO[R+E] 19/3, cKO[R+E+M] 20/3. (**M**) Comparison of IPSCs normalized to their own controls for cKO[R+E] (absolute data from *Figure 1E*) and cKO[R+E+M] (from **J**) neurons, cKO[R+E] 18/3, cKO[R+E+M] 31/3. (**N**) Observed ratio and distribution of T* values for NMDAR-EPSCs after 100,000 rounds of hierarchical bootstrap, T* = [mean(cKO[R+E])/mean(control[R+E])]/[mean(cKO[R+E+M])/mean(control[R+E+M])]; 95% confidence intervals (CI) and the probability of the null hypothesis $P_{H0}$ are indicated, n as in (**L**). (**O**) As (**N**), but for IPSCs, n as in (**M**). Data are mean ± SEM unless noted otherwise; **p<0.01, ***p<0.001 as determined by Welch's *t*-tests (**K, L**) or Mann–Whitney tests (**B, C, E, F, H, J, M**). For mEPSC and mIPSC kinetics in cKO[R+E+M] neurons, see *Figure 4—figure supplement 1*. For a workflow of the hierarchical bootstrap, see *Figure 4—figure supplement 2*.

The online version of this article includes the following figure supplement(s) for figure 4:

**Figure supplement 1.** Assessment for the kinetics of spontaneous vesicle release in RIM + ELKS + Munc13 knockout neurons.

**Figure supplement 2.** Workflow of the hierarchical bootstrap analyses.

and inhibitory (*Figure 4I–K*) synapses compared to control[R+E+M] neurons. To directly test whether the additional Munc13 ablation further reduced release compared to cKO[R+E] (*Figure 1B–E*), we normalized the PSC amplitudes measured in cKO[R+E] and cKO[R+E+M] neurons to their own controls, which are genetically identical within each mouse line except for the expression of Cre recombinase. We then compared the normalized data using standard statistical tests (*Figure 4L and M*). These analyses revealed that for both excitatory and inhibitory synapses, Munc13 knockout in addition to RIM + ELKS knockout decreased the remaining PSCs by ~40–50% compared to RIM + ELKS knockout only (*Figure 4L and M*). The design of the experiments imparts structure to the data: repeated measurements (sweeps) from one cell are more similar to each other than to measurements from another cell, and the cells from one culture batch might be more similar to each other than to those from other batches. To account for this structure in our experiments, we performed a hierarchical bootstrap (*Saravanan et al., 2020*) following the workflow in *Figure 4—figure supplement 2*. Through hierarchical resampling, we calculated 100,000 bootstrap replicates (T* values) of our test statistic T (the control-normalized PSC ratio of cKO[R+E] to cKO[R+E+M]), thus allowing us to estimate the sampling distribution of T, to calculate 95% confidence intervals, and to calculate the probability $P_{H0}$ of the null hypothesis (T* ≤ 1) given the data. For both datasets, these distributions were robustly above 1 with correspondingly low probabilities for the null hypothesis (*Figure 4N and O*). Thus, using both standard statistical tests and a hierarchical bootstrap, we found strong evidence for a robust, additional decrease in action potential-evoked release after ablation of Munc13. The remaining release in cKO[R+E+M] neurons might be due to the low amount of exon 21/22-deficient Munc13-1 that persists after conditional Munc13-1 knockout with this allele (*Banerjee et al., 2022*; *Figure 1—figure supplement 2*), to Munc13-1 that persists beyond 11 days of Cre expression, to very low levels of Munc13-3 that escaped our detection in previous studies as Munc13-3 was not deleted in the hextuple knockout mice, or to an alternative release pathway that does not depend on RIM, ELKS, and Munc13. Altogether, however, the data establish that the remaining neurotransmitter release after RIM and ELKS knockout depends at least partially on the presence of Munc13-1 and Munc13-2.

## Munc13 contributes to a remaining functional RRP after active zone disruption

Given the further reduction of synaptic transmission when Munc13 is ablated in cKO[R+E] neurons, we analyzed vesicle priming and release probability in cKO[R+E+M] neurons. The goal was to determine which release properties are controlled by Munc13 through comparison of these parameters with cKO[R+E] neurons. We assessed the functional RRP at both excitatory and inhibitory synapses through the application of hypertonic sucrose, a method that has been broadly used to evaluate correlations between vesicle docking and priming (*Imig et al., 2014*; *Rosenmund and Stevens, 1996*; *Wang et al., 2016*; *Zarebidaki et al., 2020*). We detected robust reductions of the vesicle pool assessed by this method for both AMPAR and GABA_AR-mediated responses in cKO[R+E] neurons (*Figure 5A–D*), but the functional RRP was not fully eliminated. Deletion of Munc13 on top of RIM and ELKS revealed an additional decrease, with an almost complete loss of releasable vesicles at excitatory cKO[R+E+M] synapses and an >80% reduction at inhibitory cKO[R+E+M] synapses (*Figure 5E–H*). Comparison of these two genotypes by a standard statistical approach showed that cKO[R+E+M] neurons had a significantly smaller RRP size than cKO[R+E] neurons in both synapse types (*Figure 5I and J*). The hierarchical

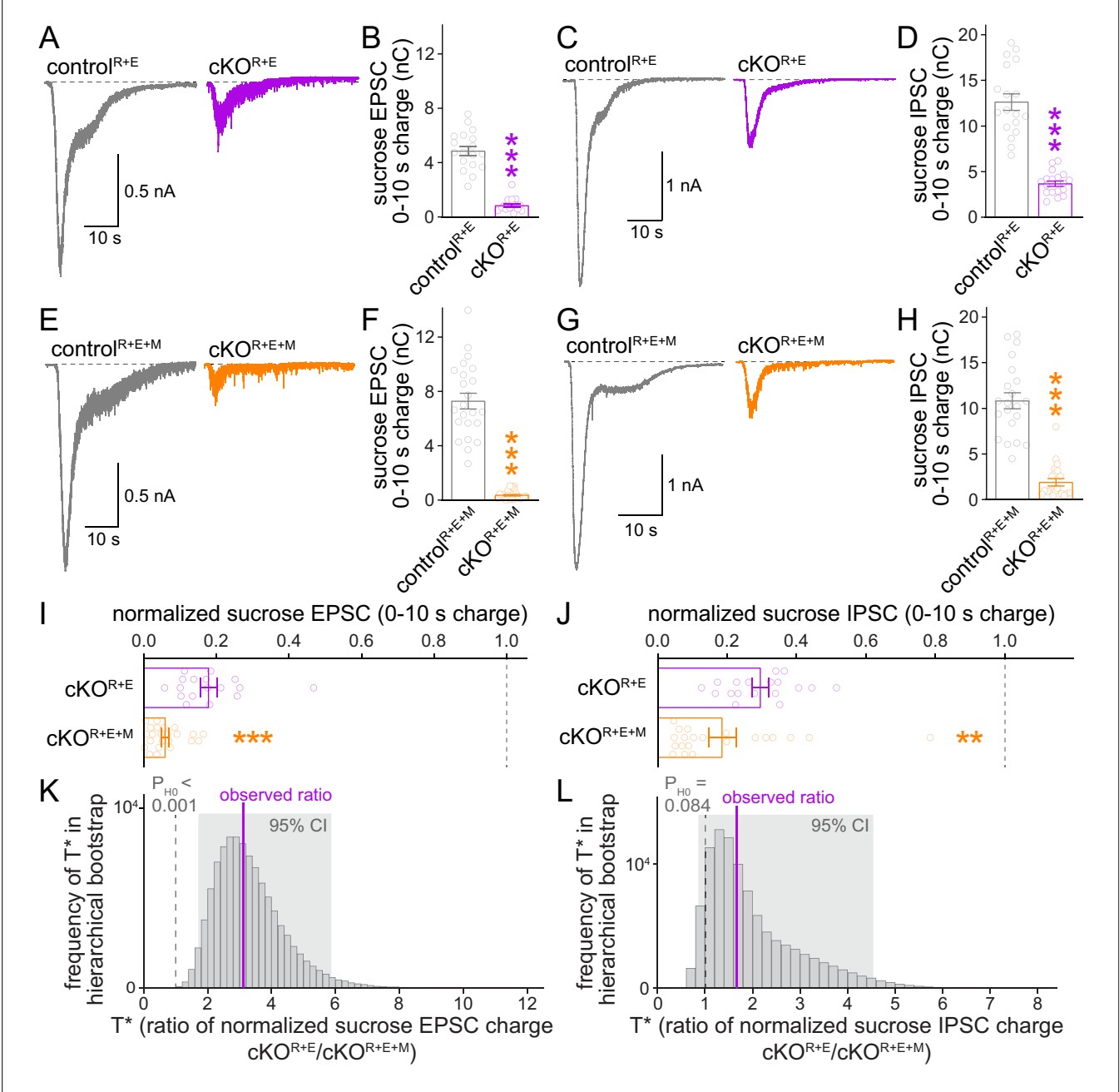

**Figure 5.** The remaining functional RRP in RIM + ELKS-deficient synapses depends at least in part on Munc13. (**A, B**) Sample traces (**A**) and quantification (**B**) of excitatory postsynaptic currents (EPSCs) triggered by hypertonic sucrose in control$^{R+E}$ and cKO$^{R+E}$ neurons, the first 10 s of the EPSC were quantified to estimate the RRP, control$^{R+E}$ 18 cells/3 cultures, cKO$^{R+E}$ 17/3. (**C, D**) As (**A, B**), but for inhibitory postsynaptic currents (IPSCs), 18/3 each. (**E–H**) As for (**A–D**), but for control$^{R+E+M}$ and cKO$^{R+E+M}$ neurons, (**F**) 23/3 each, (**H**) 21/3 each. (**I**) Comparison of EPSCs triggered by hypertonic sucrose normalized to their own controls for cKO$^{R+E}$ (absolute data from **B**) and cKO$^{R+E+M}$ (from **F**), cKO$^{R+E}$ 17/3, cKO$^{R+E+M}$ 23/3. (**J**) Comparison of IPSCs triggered by hypertonic sucrose normalized to their own controls for cKO$^{R+E}$ (absolute data from **D**) and cKO$^{R+E+M}$ (from **H**), cKO$^{R+E}$ 18/3, cKO$^{R+E+M}$ 21/3. (**K**) Observed ratio and distribution of T* values for sucrose EPSC charge after 100,000 rounds of hierarchical bootstrap, n as in (**I**). (**L**) As (**K**), but for sucrose IPSC charge, n as in (**J**). Data are mean ± SEM unless noted otherwise; **p<0.01, ***p<0.001 as determined by Mann–Whitney tests (**B, F, H, I, J**) or Welch's *t*-tests (**D**).

bootstrap analyses revealed probabilities for the null hypotheses to be true of <0.001 for excitatory synapses and of 0.084 for inhibitory synapses, respectively (**Figure 5K and L**). We conclude that the fusion competence of vesicles that remains after active zone disruption by RIM + ELKS knockout is mediated at least in part by Munc13. Because Munc13 is not active zone-anchored and docked

vesicles are barely detectable in cKO$^{R+E}$ neurons, these vesicles are undocked, but likely associated with Munc13 at some distance away from the active zone.

We finally used paired pulse ratios to monitor vesicular release probability p in control$^{R+E+M}$ and cKO$^{R+E+M}$ neurons and repeated the measurements in cKO$^{R+E}$ neurons for direct comparison (*Figure 6*). Paired pulse ratios are inversely correlated with p (*Zucker and Regehr, 2002*) and strongly increased at short interstimulus intervals after knockout of RIM and ELKS at excitatory and inhibitory synapses (*Figure 6A–D*; *Tan et al., 2022*; *Wang et al., 2016*). In cKO$^{R+E+M}$ neurons, p was also strongly decreased, illustrated by robust increases in EPSC and IPSC paired pulse ratios (*Figure 6E–H*), to an extent that is overall very similar to cKO$^{R+E}$ neurons. Comparison of genotypes through standard statistical methods or hierarchical bootstrap supported the conclusion that Munc13 knockout in addition to RIM and ELKS did not further decrease p, as effects in cKO$^{R+E}$ synapses and in cKO$^{R+E+M}$ synapses were indistinguishable (*Figure 6I–L*). Consistent with this comparison, spontaneous mEPSC and mIPSC frequencies (*Figure 6—figure supplement 1A–H*) and depression of IPSCs during stimulus trains (*Figure 6—figure supplement 1I–L*) were similar in the two genotypes. Hence, while the remaining Munc13 at cKO$^{R+E}$ synapses is sufficient to maintain a small functional RRP, it does not enhance vesicular release probability of these vesicles.

## Discussion

We found previously that the functional RRP is not fully disrupted after ablating vesicle docking by simultaneous knockout of RIM and ELKS (*Wang et al., 2016*). Here, we show that fusion of these remaining RRP vesicles depends at least in part on Munc13. Even though Munc13-1 is not active zone-anchored after RIM + ELKS ablation, knocking out Munc13 in addition decreased the remaining pool of releasable vesicles at excitatory and inhibitory hippocampal synapses. We conclude that Munc13 can render some vesicles fusogenic in the absence of RIM and ELKS. Our work adds a very strong compound knockout mutation to disrupt synaptic function, with removal of six important active zone proteins, to a growing body of literature that is based on the compound mutant approach. Beyond its relevance for mechanisms of neurotransmitter release, our work further supports that synapse formation is remarkably resilient to even massive perturbations of presynaptic function and protein composition.

### Disrupting active zones by removing redundancy

Knockout studies on active zone gene families have defined multiple roles for these proteins in the neurotransmitter release process. While some functions and mechanisms, for example, vesicle priming, strongly depend on single proteins (*Aravamudan et al., 1999*; *Augustin et al., 1999*; *Richmond et al., 1999*; *Varoqueaux et al., 2002*), other functions are redundant between members within a protein family or across protein families, and hence much more difficult to study mechanistically. This is particularly true for the scaffolding mechanisms that hold the active zone together and connect it with the plasma membrane and the vesicle cluster. These mechanisms are not well defined, and models ranging from self-assembly of complexes with defined stoichiometries to phase separation via multivalent low-affinity interactions have been proposed (*Chen et al., 2020*; *Emperador-Melero and Kaeser, 2020*; *Südhof, 2012*). Recent studies with compound mutants to remove combinations of active zone protein families started to identify the required active zone scaffolds (*Acuna et al., 2016*; *Kushibiki et al., 2019*; *Oh et al., 2021*; *Wang et al., 2016*). We found previously that simultaneous deletion of RIM and ELKS in hippocampal neurons strongly disrupts active zone protein assemblies with subsequent loss of Munc13, RIM-BP, Piccolo, and Bassoon (*Tan et al., 2022*; *Wang et al., 2016*; *Wong et al., 2018*). This loss of active zone material causes a near-complete disruption of vesicle docking and strongly impairs action potential-triggered neurotransmitter release. Unexpectedly, however, some release persists either due to the presence of remaining release-competent but non-docked vesicles or due to the rapid generation of release-competent vesicles in response to stimulation. The present study establishes that the transmitter release that remains after RIM and ELKS deletion relies at least in part on Munc13 as ablating Munc13 on top of RIM and ELKS decreased the remaining functional RRP further. The remaining release in cKO$^{R+E+M}$ neurons might be Munc13-independent, which is currently not possible to establish. Perhaps more likely, it is mediated by the small amount of remaining Munc13-1 protein in the neurons derived from the conditional mutant

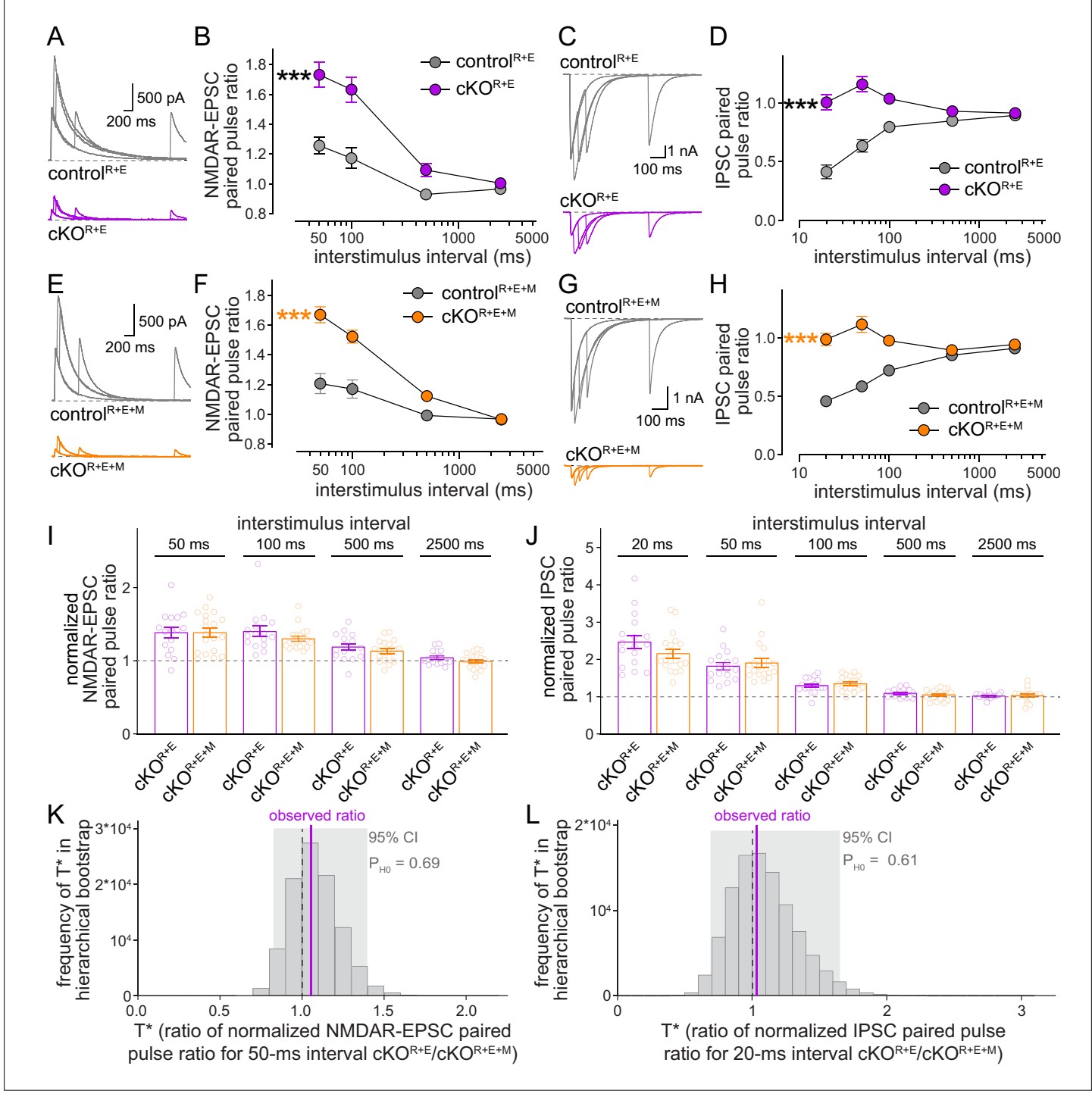

**Figure 6.** Vesicular release probability is not further impaired by combined RIM + ELKS + Munc13 knockout. (**A, B**) Sample traces (**A**) and quantification (**B**) of excitatory postsynaptic current (EPSC) paired pulse ratios in control[R+E] and cKO[R+E] neurons, control[R+E] 15 cells/3 cultures, cKO[R+E], 16/3. (**C, D**) As (**A**, **B**), but for inhibitory postsynaptic currents (IPSCs) (sample traces of 2500 ms intervals are not shown in **C** for simplicity), 17/3 each. (**E–H**) As for (**A–D**), but for control[R+E+M] and cKO[R+E+M] neurons, (**F**) 19/3 each, (**H**) 19/3 each. (**I**) Comparison of EPSC paired pulse ratios across interstimulus intervals normalized to their own controls for cKO[R+E] (absolute data from **B**) and cKO[R+E+M] (from **F**), cKO[R+E] 16/3, cKO[R+E+M] 19/3. (**J**) Comparison of IPSC paired pulse ratios across interstimulus intervals normalized to their own controls for cKO[R+E] (absolute data from **D**) and cKO[R+E+M] (from **H**), cKO[R+E] 17/3, cKO[R+E+M] 19/3. (**K**) Observed ratio and distribution of T* values for EPSC paired pulse ratios at 50 ms interstimulus intervals after 100,000 rounds of hierarchical bootstrap, n as in (**I**). (**L**) As (**K**), but for IPSC paired pulse ratios at 20 ms interstimulus intervals, n as in (**J**). Data are mean ± SEM unless noted otherwise; ***p<0.001 as determined by two-way ANOVA followed by Bonferroni's multiple-comparisons post-hoc tests (**B, D, F, H**) or by Mann–Whitney tests

*Figure 6 continued on next page*

*Figure 6 continued*

(**I, J**). For miniature EPSCs (mEPSCs) and miniature IPSCs (mIPSCs) in cKO$^{R+E}$ neurons, and for IPSCs evoked by stimulus trains in cKO$^{R+E}$ and cKO$^{R+E+M}$ neurons, see *Figure 6—figure supplement 1*.

The online version of this article includes the following figure supplement(s) for figure 6:

**Figure supplement 1.** Additional analyses of spontaneous vesicle release and inhibitory postsynaptic currents (IPSCs) evoked by stimulus trains in RIM + ELKS and RIM + ELKS + Munc13 knockout neurons.

mice used here (*Banerjee et al., 2022*), which is not seen with the constitutive Munc13-1 knockout mouse line (*Augustin et al., 1999*; *Man et al., 2015*). We further found that synapse structure per se is resilient to this major genetic and functional perturbation as synapses formed at normal densities and showed overall near-normal ultrastructure despite the knocking out of six important active zone genes in the cultured neurons. Our work adds to a growing body of data demonstrating that neurotransmitter release, presynaptic Ca$^{2+}$ entry, and active zone scaffolding are all dispensable for the formation of prominent types of central nervous system synapses (*Held et al., 2020*; *Sando et al., 2017*; *Sigler et al., 2017*; *Verhage et al., 2000*).

One way to interpret our data is to compare them with properties of fusion in other secretory pathways. Synaptic vesicle exocytosis after RIM and ELKS knockout has resemblance with secretion from chromaffin cells (*Neher, 2018*; *Wang et al., 2016*). In these cells, the release-ready pool of vesicles depends on Munc13, has relatively slower kinetics compared to synapses, but appears to not rely on the sequences of Munc13 that interact with RIM or on active zone scaffolds more generally (*Betz et al., 2001*; *Man et al., 2015*; *Neher, 2018*; *Stevens et al., 2005*). The similarities of the remaining fusion after disrupting active zone assembly by knockout of RIM and ELKS are striking in that the release kinetics appear slowed down (*Wang et al., 2016*), the remaining fusion depends at least in part on Munc13, and docking and the functional RRP are not fully correlated. Some of these similarities are also reminiscent of recent studies in *C. elegans*, where vesicle priming does not rely on the interaction of unc-13/Munc13 with unc-10/RIM, and possibly unc-10/RIM itself (*Liu et al., 2019*). Altogether, these studies and our work indicate that disrupting the active zone through RIM + ELKS knockout renders synaptic vesicle exocytosis similar to chromaffin cell secretion. Notably, there are specific vertebrate synapses with similar properties. For example, at hippocampal mossy fiber synapses, strong depolarizations lead to RRP estimates that are larger than the number of docked vesicles (*Maus et al., 2020*), supporting the model that RRP vesicles can be rapidly generated and released during such stimuli, perhaps similar to the remaining RRP at cKO$^{R+E}$ synapses.

## Can Munc13 prime vesicles in the absence of RIM?

Our data reveal that the transmitter release that remains after active zone disruption upon RIM and ELKS deletion depends at least in part on Munc13. Thus, Munc13 might render some vesicles fusion-competent in the absence of RIM and when Munc13 is not anchored at the active zone. This mechanism, however, is inefficient as it only maintains a fraction of the functional RRP of a wildtype synapse. An alternative or complementary model is that fusion competence is rapidly generated and immediately followed by exocytosis during pool-depleting stimuli. We recently established that the functional RRP after RIM + ELKS knockout can be further boosted by re-expressing RIM zinc finger domains without enhancing vesicle docking (*Tan et al., 2022*). When the RIM zinc finger domain was expressed in RIM + ELKS knockout neurons, it co-localized with the vesicle cluster, it was not concentrated at the active zone, and it recruited Munc13 to the vesicle cluster. Hence, Munc13 can enhance vesicle fusogenicity through association with non-docked vesicles, at least when this association is generated artificially. Here, we show that the remaining fusion in RIM + ELKS knockouts depends at least in part on Munc13 (*Figures 4 and 5*), but that Munc13-1 is barely detectable at active zones, while some Munc13 is present at synapses. These data indicate that endogenous Munc13 can be near vesicles and enhance their fusogenicity even if Munc13 is not active zone-anchored. Hence, Munc13 on non-docked vesicles might mediate the remaining fusion through generation of a pool of vesicles that can be rapidly primed and released upon stimulation. Altogether, a model arises that the rate-limiting step for generation of a functional RRP is the presynaptic recruitment of Munc13, even if Munc13 is not fully anchored and activated at the active zone. Upon stimulation, roles of Munc13 in SNARE complex assembly and fusion can be executed quickly.

Previous work established that RIM recruits Munc13 to active zones and activates it. This mechanism operates through binding of the RIM zinc finger to Munc13 $C_2A$ domains, which is necessary for rendering Munc13 monomeric and active in fusion (*Andrews-Zwilling et al., 2006*; *Betz et al., 2001*; *Brockmann et al., 2020*; *Camacho et al., 2017*; *Deng et al., 2011*; *Dulubova et al., 2005*; *Lu et al., 2006*). In the experiments presented here, the neurotransmitter release remaining in the absence of RIM and ELKS depends at least partially on Munc13. This indicates that not all priming requires the RIM-mediated activation mechanism of Munc13, consistent with previous observations that many but not all RRP vesicles are lost after RIM knockout (*Han et al., 2011*; *Han et al., 2015*; *Kaeser et al., 2011*; *Kaeser et al., 2012*). An alternative mechanism could operate via bMunc13-2 (which lacks the $C_2A$-domain that binds to RIM) and ELKS (*Kawabe et al., 2017*). While ELKS provides a Munc13-recruitment and priming mechanism that is independent of RIM, most ELKS is also removed in the RIM + ELKS knockout neurons (*Kaeser et al., 2009*; *Liu et al., 2014*; *Wang et al., 2016*), and only a very small subset of synapses relies on bMunc13-2 (*Kawabe et al., 2017*). Hence, this mechanism might be insufficient to explain the remaining release from RIM + ELKS knockout neurons. An alternative and perhaps more likely explanation is that not all $C_2A$-domain-containing Munc13 requires RIM. Monomeric, active Munc13 is in equilibrium with dimeric, inactive Munc13, and RIM shifts the equilibrium to the active form. Even in the absence of RIM, some Munc13 will be monomeric and available for assembling SNARE complexes, accounting for the vesicular exocytosis that remains in RIM + ELKS knockout neurons. Finally, this mechanism may not be restricted to docked vesicles, but vesicles associated with Munc13 may be amenable to release, explaining why some release persists in RIM + ELKS mutants despite the loss of active zone-anchored Munc13 and of a strong reduction in docked vesicles.

## Materials and methods

**Key resources table**

| Reagent type (species) or resource | Designation | Source or reference | Identifiers | Additional information |
|---|---|---|---|---|
| Genetic reagent (*Mus musculus*) | *Rims1*$^{tm3Sud}$/J (RIM1αβ$^{fl/fl}$) | *Kaeser et al., 2008* | RRID:IMSR_JAX:015832 | |
| Genetic reagent (*M. musculus*) | *Rims2*$^{tm1.1Sud}$/J (RIM2αβγ$^{fl/fl}$) | *Kaeser et al., 2011* | RRID:IMSR_JAX:015833 | |
| Genetic reagent (*M. musculus*) | *Erc1*$^{tm2.1Sud}$/J (ELKS1α$^{fl/fl}$) | *Liu et al., 2014* | RRID:IMSR_JAX:015830 | |
| Genetic reagent (*M. musculus*) | *Erc2*$^{tm1.2Sud}$/J (ELKS2α$^{fl/fl}$) | *Kaeser et al., 2009* | RRID:IMSR_JAX:015831 | |
| Genetic reagent (*M. musculus*) | *Unc13a*$^{tm1.1Bros}$ (Munc13-1$^{fl/fl}$) | *Banerjee et al., 2022* | MGI:7276178 | |
| Genetic reagent (*M. musculus*) | *Unc13b*$^{tm1Rmnd}$ (Munc13-2$^{-/-}$) | *Varoqueaux et al., 2002* | RRID:MGI:2449706 | |
| Genetic reagent (*M. musculus*) | *Unc13c*$^{tm1Bros}$ (Munc13-3$^{-/-}$) | *Augustin et al., 2001* | RRID:MGI:2449467 | |
| Cell line (*Homo sapiens*) | HEK293T cells | ATCC | Cat# CRL-3216; RRID:CVCL_0063 | |
| Recombinant DNA reagent | pFSW EGFP Cre | *Liu et al., 2014* | pHN131014; lab plasmid code (LPC): p009 | |
| Recombinant DNA reagent | pFSW EGFP ΔCre | *Liu et al., 2014* | pHN131015; LPC: p010 | |
| Antibody | Anti-RIM (rabbit polyclonal) | SySy | Cat# 140003; RRID:AB_887774; lab antibody code (LAC): A58 | Immunofluorescence (IF) (1:500) |
| Antibody | Anti-PSD-95 (mouse monoclonal) | NeuroMab | Cat# 73-028; RRID:AB_10698024; LAC: A149 | IF (1:500) |

*Continued on next page*

*Continued*

| Reagent type (species) or resource | Designation | Source or reference | Identifiers | Additional information |
|---|---|---|---|---|
| Antibody | Anti- Synaptophysin (guinea pig polyclonal) | SySy | Cat# 101004; RRID:AB_1210382; LAC: A106 | IF (1:500) |
| Antibody | Anti-Munc13-1 (rabbit polyclonal) | SySy | Cat# 126103; RRID:AB_887733; LAC: A72 | IF (1:500); Western blot (WB) (1:1000) |
| Antibody | Anti-Synapsin-1 (mouse monoclonal) | SySy | Cat# 106001; RRID:AB_2617071; LAC: A57 | WB (1:4000) |
| Antibody | Anti-rabbit Alexa Fluor 488 (goat polyclonal) | Thermo Fisher | Cat# A-11034; RRID:AB_2576217; LAC: S5 | IF (1:200) |
| Antibody | Anti-mouse IgG2a Alexa Fluor 555 (goat polyclonal) | Thermo Fisher | Cat# A-21137; RRID:AB_2535776; LAC: S20 | IF (1:200) |
| Antibody | Anti-guinea pig Alexa Fluor 633 (goat polyclonal) | Thermo Fisher | Cat# A-21105; RRID:AB_2535757; LAC: S34 | IF (1:500) |
| Antibody | Anti-mouse peroxidase-conjugated (goat polyclonal) | MP Biologicals | Cat# 0855550; RRID:AB_2334540; LAC: S52 | WB (1:10000) |
| Antibody | Anti-rabbit peroxidase-conjugated (goat polyclonal) | MP Biologicals | Cat# 0855676; RRID:AB_2334589; LAC: S53 | WB (1:10000) |
| Software, algorithm | MATLAB | MathWorks | RRID:SCR_001622; https://www.mathworks.com/products/matlab.html | |
| Software, algorithm | Prism | GraphPad | RRID:SCR_002798; https://www.graphpad.com/scientific-software/prism | |
| Software, algorithm | Fiji/ImageJ | NIH | RRID:SCR_002285; https://imagej.net/software/fiji/downloads | |
| Software, algorithm | pClamp | Molecular Devices | RRID:SCR_011323; https://www.moleculardevices.com/products/software/pclamp.html | |
| Software, algorithm | MATLAB code for object recognition and analysis of 2D images | This paper | https://doi.org/10.5281/zenodo.6388196 | This code was adapted for 2D images from a previously generated code (*Liu et al., 2018*; *Liu et al., 2022*) and is freely accessible at zenodo.org |
| Software, algorithm | MATLAB code for hierarchical bootstrap | This paper | https://github.com/kaeserlab/Hierarchical_Bootstrap_Analysis_RB; *Born, 2022* | This code was generated for this paper and is freely accessible at github.com |

## Mouse lines

The quadruple homozygote floxed mice for *Rims1* (to remove RIM1α and RIM1β, RRID:IMSR_JAX:015832, *Kaeser et al., 2008*), *Rims2* (to remove RIM2α, RIM2β, and RIM2γ, RRID:IMSR_JAX:015833, *Kaeser et al., 2011*), *Erc1* (to remove ELKS1α, RRID:IMSR_JAX:015830, *Liu et al., 2014*), and *Erc2* (to remove ELKS2α, RRID:IMSR_JAX:015831, *Kaeser et al., 2009*) were previously described (*Wang et al., 2016*). Exon 6 (E6) or 26 (E26) were flanked by loxP sites in the *Rims1* or *Rims2* floxed mice, respectively. Exons 2 (E2) and 3 (E3) were flanked by loxP sites in the *Erc1* floxed mice, and exon 3 (E3) was flanked by loxP sites in the *Erc2* floxed mice. Floxed *Unc13a* mice (Munc13-1, Exon 21 [E21] flanked by loxP sites, *Unc13a*^tm1.1Bros^, MGI:7276178, *Banerjee et al., 2022*) were crossed to constitutive knockout mice for *Unc13b* (Munc13-2, *Unc13b*^tm1Rmnd^, RRID_MGI:2449706, *Varoqueaux et al., 2002*) and *Unc13c* (Munc13-3, *Unc13c*^tm1Bros^, RRID_MGI:2449467, *Augustin et al., 2001*) to produce Munc13 triple homozygote mice. Mice for simultaneous ablation of RIM1, RIM2, ELKS1, ELKS2, Munc13-1, and Munc13-2 were generated by crossing the corresponding conditional (RIM1αβ, RIM2αβγ, ELKS1α, ELKS2α, and Munc13-1) and constitutive (Munc13-2) knockout alleles to homozygosity. All animal experiments were approved by the Harvard University Animal Care and Use Committee (protocol number IS00000049).

## Cell lines, primary neuronal culture, and lentiviral infection

Primary mouse hippocampal cultures were generated from newborn pups within 24 hr after birth as described (*Held et al., 2020*; *Tan et al., 2022*; *Wang et al., 2016*); cells from mice of both sexes were mixed. Mice were anesthetized by hypothermia and the hippocampus was dissected out. Cells were dissociated and plated onto glass coverslips in tissue culture medium composed of Minimum Essential Medium (MEM) with 10% Fetal Select bovine serum (Atlas Biologicals FS-0500-AD), 2 mM L-glutamine, and 25 µg/mL insulin, 0.1 mg/mL transferrin, 0.5% glucose, and 0.02% NaHCO$_3$. Cultures were maintained in a 37°C tissue culture incubator, and after ~24 hr the plating medium was exchanged with growth medium composed of MEM with 5% Fetal Select bovine serum, 2% B-27 supplement (Thermo Fisher 17504044), 0.5 mM L-glutamine, 0.1 mg/mL transferrin, 0.5% glucose, and 0.02% NaHCO$_3$. At DIV3, depending on growth, 50 or 75% of the medium were exchanged with growth medium supplemented with 4 µM cytosine β-D-arabinofuranoside (AraC). Cultured neurons were transduced with lentiviruses produced in HEK293T cells (CRL-3216, RRID:CVCL_0063, immortalized human cell line of female origin, purchased mycoplasma-free) by Ca$^{2+}$ phosphate transfection; viral transduction was at DIV5 unless noted otherwise. These lentiviruses expressed EGFP-tagged Cre recombinase (to generate cKO neurons) or a truncated, enzymatically inactive EGFP-tagged Cre protein (ΔCre, to generate control neurons). Expression in lentiviral constructs was driven by the human Synapsin-1 promoter to restrict expression to neurons (*Liu et al., 2014*; *Wang et al., 2016*). Analyses were performed at DIV16–19.

## Electrophysiology

Electrophysiological recordings in cultured hippocampal neurons were performed as described (*Held et al., 2020*; *Tan et al., 2022*; *Wang et al., 2016*) at DIV16–19. The extracellular solution contained (in mM) 140 NaCl, 5 KCl, 2 MgCl$_2$, 1.5 CaCl$_2$, 10 glucose, 10 HEPES-NaOH (pH 7.4, ~300 mOsm). To avoid network activity induced by AMPAR activation, NMDAR-mediated excitatory postsynaptic currents (NMDAR-EPSCs) were measured to assess action potential-triggered excitatory transmission. For NMDAR-EPSCs, 6-Cyano-7-nitroquinoxaline-2,3-dione (CNQX, 20 µM) and picrotoxin (PTX, 50 µM) were present in the extracellular solution. Electrically evoked inhibitory postsynaptic currents (IPSCs) were recorded in the presence of D-amino-5-phosphonopentanoic acid (D-APV, 50 µM) and CNQX (20 µM) in the extracellular solution. Recordings were performed at room temperature (20–24°C). Action potentials were elicited with a bipolar focal stimulation electrode fabricated from nichrome wire. Paired pulse ratios were calculated as the amplitude of the second PSC divided by the amplitude of the first at each interval from the average of three to four sweeps per cell and interval. The baseline value for the second PSC was taken immediately after the second stimulus artifact. For analysis of action potential trains (50 stimuli at 10 Hz), the baseline value of each IPSC within the train was taken immediately after the corresponding stimulus artifact. For AMPAR-mediated transmission (mEPSC and sucrose-induced EPSCs), TTX (1 µM), PTX (50 µM), and D-APV (50 µM) were added to the extracellular solution. For mIPSC and sucrose-induced IPSC recordings, TTX (1 µM), CNQX (20 µM), and D-APV (50 µM) were added to the extracellular solution. The RRP was estimated by application of 500 mM sucrose in extracellular solution applied via a microinjector syringe pump for 10 s at a rate of 10 µL/min through a tip with an inner diameter of 250 µm. mEPSCs and mIPSCs were identified with a template search followed by manual confirmation by an experimenter, and their frequencies and amplitudes were assessed during a 100 s recording time window after reaching a stable baseline (>3 min after break-in). Glass pipettes were pulled at 2–5 MΩ and filled with intracellular solutions containing (in mM) for EPSC recordings: 120 Cs-methanesulfonate, 2 MgCl$_2$, 10 EGTA, 4 Na$_2$-ATP, 1 Na-GTP, 4 QX314-Cl, 10 HEPES-CsOH (pH 7.4, ~300 mOsm); and for IPSC recordings: 40 CsCl, 90 K-gluconate, 1.8 NaCl, 1.7 MgCl$_2$, 3.5 KCl, 0.05 EGTA, 2 Mg-ATP, 0.4 Na$_2$-GTP, 10 phosphocreatine, 4 QX314-Cl, 10 HEPES-CsOH (pH 7.2, ~300 mOsm). Cells were held at +40 mV for NMDAR-EPSC recordings and at –70 mV for evoked IPSC, mEPSC, mIPSC, sucrose EPSC, and sucrose IPSC recordings. Access resistance was monitored and compensated to 3–5 MΩ, and cells were discarded if the uncompensated access exceeded 15 MΩ. Data were acquired at 5 kHz and lowpass filtered at 2 kHz with an Axon 700B Multiclamp amplifier and digitized with a Digidata 1440A digitizer. Data acquisition and analyses were done using pClamp10. For electrophysiological experiments, the experimenter was blind to the genotype throughout data acquisition and analyses.

## STED and confocal imaging

Light microscopic analyses were in essence performed as previously described (*Emperador-Melero et al., 2021b*; *Emperador-Melero et al., 2021a*; *Held et al., 2020*; *Tan et al., 2022*; *Wong et al., 2018*). Neurons were cultured on 0.17-mm-thick 12-mm-diameter coverslips. At DIV16–18, cultured neurons were washed two times with warm PBS and fixed in 4% PFA in PBS for 10 min. After fixation, coverslips were rinsed twice in PBS, then permeabilized in PBS + 0.1% Triton X-100 + 3% BSA (TBP) for 1 hr. Primary antibodies were diluted in TBP and stained for 24–48 hr at 4°C. The following primary antibodies were used: rabbit anti-RIM1 (1:500, RRID:AB_887774, A58), rabbit anti-Munc13-1 (1:500, RRID:AB_887733, A72), guinea pig anti-Synaptophysin (1:500, RRID:AB_1210382, A106), mouse anti-PSD-95 (1:500, RRID:AB_10698024, A149). After primary antibody staining, coverslips were rinsed twice and washed 3–4 times for 5 min in TBP. Alexa Fluor 488 (anti-rabbit, RRID:AB_2576217, S5), 555 (anti-mouse IgG2a, RRID:AB_1500824, S20), and 633 (anti-guinea pig, RRID:AB_2535757, S34) conjugated antibodies were used as secondary antibodies at 1:200 (Alexa Fluor 488 and 555) or 1:500 (Alexa Fluor 633) dilution in TBP, incubated for 24–48 hr at 4°C followed by rinsing two times and washing 3–4 times 5 min in TBP. Stained coverslips were post-fixed for 10 min with 4% PFA in PBS, rinsed two times in PBS + 50 mM glycine, and once in deionized water, air-dried, and mounted on glass slides. STED images were acquired with a Leica SP8 Confocal/STED 3X microscope with an oil immersion 1.44 numerical aperture 100x objective and gated detectors as described before (*Emperador-Melero et al., 2021a*; *Held et al., 2020*; *Tan et al., 2022*; *Wong et al., 2018*). Images of 46.51 × 46.51 µm$^2$ areas were scanned at a pixel density of 4096 × 4096 (11.4 nm/pixel). Alexa Fluor 633, Alexa Fluor 555, and Alexa Fluor 488 were excited with 633 nm, 555 nm, and 488 nm using a white light laser at 2–5% of 1.5 mW laser power. The Alexa Fluor 633 channel was acquired first in confocal mode using 2× frame averaging. Subsequently, Alexa Fluor 555 and Alexa Fluor 488 channels were acquired in both confocal and STED modes. Alexa Fluor 555 and 488 channels in STED mode were depleted with 660 nm (50% of max power, 30% axial depletion) and 592 nm (80% of max power, 30% axial depletion) depletion lasers, respectively. Line accumulation (2–10×) and frame averaging (2×) were applied during STED scanning. Identical settings were applied to all samples within an experiment. Synapses within STED images were selected in side-view, defined as synapses that contained a synaptic vesicle cluster labeled with Synaptophysin and associated with an elongated PSD-95 structure along the edge of the vesicle cluster as described (*Emperador-Melero et al., 2021b*; *Emperador-Melero et al., 2021a*; *Held et al., 2020*; *de Jong et al., 2018*; *Nyitrai et al., 2020*; *Tan et al., 2022*; *Wong et al., 2018*). For intensity profile analyses, side-view synapses were selected using only the PSD-95 signal and the vesicle signal for all experiments by an experimenter blind to the protein of interest. A region of interest (ROI) was manually drawn around the PSD-95 signal and fit with an ellipse to determine the center position and orientation. A ~1200-nm-long, 200-nm-wide rectangle was then positioned perpendicular to and across the center of the elongated PSD-95 structure. Intensity profiles from –400 nm (presynaptic) to +200 nm (postsynaptic) relative to the center of the PSD-95 signal were obtained for all three channels within this ROI. To align individual profiles, the PSD-95 signal only was smoothened using a moving average of 5 pixels, and the smoothened signal was used to define the peak position of PSD-95. All three channels (vesicle marker, test protein, and smoothened PSD-95) were then aligned to the PSD-95 peak position and averaged across images for line profile plots. Peak values for each line profile were determined independent of peak position and used to generate the plots of peak levels. For *Figure 1—figure supplement 3A, B, D, and E* and *Figure 2—figure supplement 1C and D*, Munc13-1 levels were analyzed using Synaptophysin to define ROIs in the confocal images with ImageJ. For *Figure 2L–O* and *Figure 2—figure supplement 2A–E*, ROI selection was performed using an adapted custom-written code to perform automatic two-dimensional segmentation (*Emperador-Melero et al., 2021a*; *Held et al., 2020*; *Liu et al., 2018*; *Liu et al., 2022*); the code was deposited to Zenodo at https://doi.org/10.5281/zenodo.6388196. After Synaptophysin object detection, the density, intensity, and area of these objects were quantified (*Figure 2L–O*, *Figure 2—figure supplement 2A–D*). In *Figure 2—figure supplement 2E*, the Synaptophysin objects (confocal) that exceeded the overlap threshold of 0% with PSD-95 objects (STED) were included in the quantification. Analyses were performed on raw images without background subtraction, and adjustments were done identically across experimental conditions. Representative images were brightness- and

contrast-adjusted to facilitate inspection, and these adjustments were made identically for images within an experiment. The experimenter was blind to the condition/genotype for image acquisition and analyses.

## High-pressure freezing and electron microscopy

Neurons were cultured on sapphire coverslips (6 mm diameter) coated with matrigel. At DIV16–18, cultured neurons were frozen using a Leica EM ICE high-pressure freezer in extracellular solution containing (in mM) 140 NaCl, 5 KCl, 2 CaCl$_2$, 2 MgCl$_2$, 10 HEPES-NaOH (pH 7.4), 10 glucose (~300 mOsm), with PTX (50 µM), CNQX (20 µM), and D-AP5 (50 µM) added. After freezing, samples were first freeze-substituted (AFS2, Leica) in anhydrous acetone containing 1% glutaraldehyde, 1% osmium tetroxide, and 1% water. The process of freeze substitution was as follows: –90°C for 5 hr, 5°C per hr to –20°C, –20°C for 12 hr, and 10°C per hr to 20°C. Following freeze substitution, samples were Epon-infiltrated and baked for 48 hr at 60°C followed by 80°C overnight before sectioning at 50 nm. For ultrathin sectioning, the sapphire coverslip was removed from the resin block by plunging the sample first in liquid nitrogen and followed by warm water several times until the sapphire was completely detached. The resin block containing the neurons was then divided into four pieces, and one piece was mounted for sectioning. Ultra-thin sectioning was performed on a Leica EM UC7 ultramicrotome, and the 50 nm sections were collected on a nickel slot grid (2 × 1 mm) with a carbon-coated formvar support film. The samples were counterstained by incubating the grids with 2% lead acetate solution for 10 s, followed by rinsing with distilled water. Images were taken with a transmission electron microscope (JEOL 1200 EX at 80 kV accelerating voltage) and processed with ImageJ. The total number of vesicles, the number of docked vesicles per synapse profile, the area of the presynaptic bouton, and the length of the PSD were analyzed in each section. Docked vesicles were defined as vesicles for which the electron densities of the vesicular membrane and the presynaptic plasma membrane merged such that the two membranes were not separated by less electron-dense space. Bouton size was calculated from the measured perimeter of each synapse. Experiments and analyses were performed by an experimenter blind to the genotype.

## Western blotting

At DIV15–19, cultured neurons were harvested in 20 µL 1× SDS buffer per coverslip and run on standard SDS-PAGE gels followed by transfer to nitrocellulose membranes. Membranes were blocked in filtered 10% nonfat milk/5% goat serum for 1 hr at room temperature and incubated with primary antibodies (rabbit anti-Munc13-1, 1:1000, RRID:AB_887733, A72; mouse anti-Synapsin-1, 1:4000, RRID:AB_2617071, A57) in 5% nonfat milk/2.5% goat serum overnight at 4°C, and HRP-conjugated secondary antibodies (1:10,000, anti-mouse, RRID:AB_2334540; anti-rabbit, RRID:AB_2334589) were used. Western blotting was repeated 3–8 times per genotype from selected cultures used for electrophysiology, immunostaining, and electron microscopy. For illustration in figures, images were adjusted for brightness and contrast to facilitate visual inspection, and the same adjustments were used for the entire scan.

## Statistics

Standard statistical tests were performed using GraphPad Prism 9; hierarchical bootstrap was performed using MATLAB. Data are displayed as mean ± SEM unless noted otherwise, and significance of standard tests is presented as *p<0.05, **p<0.01, and ***p<0.001. Sample sizes were determined based on previous studies, and no statistical methods were used to predetermine sample size. No outliers were excluded. Parametric tests were used for normally distributed data (assessed by Shapiro–Wilk tests) or when sample size was n ≥ 30. Unpaired two-tailed Student's *t*-tests were used for datasets with equal variance, or Welch's unequal variances *t*-tests for datasets with unequal variance. For non-normally distributed data, Mann–Whitney tests or Kruskal–Wallis tests followed by Dunn's multiple-comparisons post-hoc tests were used. For paired pulse ratios, two-way ANOVA tests with Bonferroni's post-hoc tests were used. For STED side-view analyses, two-way ANOVA tests with Bonferroni's post-hoc tests were used on a 200-nm-window centered around the active zone peak.

The hierarchical bootstrap analyses (*Saravanan et al., 2020*) were performed using a custom-written MATLAB code (https://github.com/kaeserlab/Hierarchical_Bootstrap_Analysis_RB; *Born, 2022*). This method is similar to other hierarchical statistical approaches, such as repeated-measures ANOVA and linear mixed-effects models; however, it has the added advantage of not making any distributional assumptions about the underlying data and allows for unequal variance among the groups. To test whether knockout of Munc13 caused additional decrease for electrically or sucrose-evoked PSCs, or an increase in paired pulse ratios, the test statistic, T, was defined as:

$$T = [mean(cKO^{R+E})/mean(control^{R+E})]/[mean(cKO^{R+E+M})/mean(control^{R+E+M})]$$

The null value for T is 1, and the alternative hypothesis is T > 1 for PSCs and T < 1 for paired pulse ratios. The sampling distribution of T was estimated by resampling with replacement from the raw data while preserving the hierarchical relationships created by the design of the experiment. For each bootstrap iteration, group identity (control$^{R+E}$, cKO$^{R+E}$, control$^{R+E+M}$, or cKO$^{R+E+M}$) was preserved. Resampling was done at three nested levels: batch of culture, cell, and sweep (except for sucrose-evoked release for which only one sweep was recorded). After each bootstrap iteration for the four groups, a bootstrap replicate of T called T* was calculated using the above formula. The procedure was repeated 100,000 times, producing an estimate of the sampling distribution of T, which was plotted in the frequency histograms. Based on the distribution of the 100,000 T* values, we calculated 95% confidence intervals using the percentile method (*Efron and Tibshirani, 1994*). In addition, we calculated the probability of the null hypothesis ($P_{H0}$) given our data for action potential- and sucrose-evoked PSCs as

$$P_{H0} = \#\left\{T^* \leq 1\right\}/100,000,$$

and for paired pulse ratios as:

$$P_{H0} = \#\left\{T^* \geq 1\right\}/100,000.$$

Note that this is not the traditional p-value calculated with standard statistical tests, which is the probability of obtaining a result as extreme or more extreme assuming the null hypothesis to be true. Rather, the metric computed above is a more direct measure of a given hypothesis being true – in this case, the null hypothesis – given the measured data points (*Saravanan et al., 2020*). For all datasets, sample sizes and the specific tests used are stated in the figure legends.

## Materials, data, and code availability

Plasmids used for this study will be shared upon request. Mouse lines will be shared upon request within the limits of the respective material transfer agreements. Analyses codes have been deposited to Zenodo and GitHub and are publicly available as listed in the Key Resources Table. All data generated or analyzed in this study, including individual data points, are included in the figures. Source data files for *Figure 1—figure supplement 3*, *Figure 2—figure supplement 1*, and *Figure 2—figure supplement 2* are provided, and a source data table that contains all means, errors, statistical tests, and p-values is also included. Plasmids and mice should be requested from the corresponding author (kaeser@hms.harvard.edu).

## Acknowledgements

We thank J Wang and V Charles for technical support, all members of the Kaeser laboratory for insightful discussions and feedback, M Verhage and J Broeke for a MATLAB macro to analyze electron microscopic images, C Ma for advice on data analyses, the Harvard Biostatistics Student Consulting Center for feedback on statistical testing methods, and C Liu for adapting a 3D image analysis code for 2D images. CQ is currently a graduate student at Peking University. This work was supported by grants from the NIH (R01MH113349 and R01NS083898 to PSK) and the German Research Foundation (EXC 2067/1-390729940 to NB). We acknowledge the Neurobiology Imaging Facility (supported by a P30 Core Center Grant P30NS072030) and the Electron Microscopy Facility at Harvard Medical School.

# Additional information

## Competing interests
Nils Brose: Reviewing editor, *eLife*. The other authors declare that no competing interests exist.

## Funding

| Funder | Grant reference number | Author |
| --- | --- | --- |
| National Institute of Mental Health | MH113349 | Pascal S Kaeser |
| National Institute of Neurological Disorders and Stroke | NS083898 | Pascal S Kaeser |
| Harvard Medical School | | Pascal S Kaeser |
| Max Planck Institute for Multidisciplinary Sciences | open access funding | Cordelia Imig Nils Brose |
| German Research Foundation | EXC 2067/1-390729940 | Nils Brose |

The funders had no role in study design, data collection and interpretation, or the decision to submit the work for publication.

## Author contributions
Chao Tan, Conceptualization, Resources, Formal analysis, Investigation, Visualization, Methodology, Writing - original draft, Writing - review and editing; Giovanni de Nola, Formal analysis, Investigation, Methodology, Writing - review and editing; Claire Qiao, Resources, Investigation, Writing - review and editing; Cordelia Imig, Resources, Writing - review and editing; Richard T Born, Formal analysis, Methodology, Writing - review and editing; Nils Brose, Conceptualization, Resources, Writing - review and editing; Pascal S Kaeser, Conceptualization, Formal analysis, Supervision, Funding acquisition, Visualization, Writing - original draft, Project administration, Writing - review and editing

## Author ORCIDs
Chao Tan http://orcid.org/0000-0003-3787-0336
Claire Qiao http://orcid.org/0000-0003-2084-2478
Cordelia Imig http://orcid.org/0000-0001-7351-8706
Richard T Born http://orcid.org/0000-0003-4360-427X
Pascal S Kaeser http://orcid.org/0000-0002-1558-1958

## Ethics
This study was performed in strict accordance with the recommendations in the Guide for the Care and Use of Laboratory Animals of the National Institutes of Health. All animal experiments were approved by the Harvard University Animal Care and Use Committee (protocol number IS00000049).

## Decision letter and Author response
Decision letter https://doi.org/10.7554/eLife.79077.sa1
Author response https://doi.org/10.7554/eLife.79077.sa2

# Additional files

## Supplementary files
• MDAR checklist

• Source data 1. Numerical data for all analyses shown in the figures. Means, SEMs, statistical tests and p-values for *Figures 1–6* and figure supplements.

## Data availability

All data generated or analyzed in this study, including individual data points, are included in the figures. Source data files for Figure 1—figure supplement 3, Figure 2—figure supplement 1 and Figure 2—figure supplement 2 are provided, and a source data table that contains all means, errors, statistical tests and p-values is also included.

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
