## [Editor Report]

Tan and colleagues studied synaptic transmission, presynaptic protein levels, and synaptic ultra-structure in hippocampal cultures of mice lacking the key active-zone proteins RIM (1, 2), ELKS (1, 2), and Munc13 (1, 2). Compared to cultures lacking only RIM and ELKS, additional deletion of Munc13 results in a further decrease of synaptic Munc13-1 levels, a similar reduction of the number of docked synaptic vesicles, and a more pronounced decrease of the readily releasable vesicles. The results support the conclusion of the nonredundant role of Munc13 in synaptic vesicle priming. Overall, this study reinforces the notion that synapse formation is a remarkably resilient process that occurs even under strong perturbation of presynaptic function.

---

## [Decision Letter]

**Decision letter after peer review:**

Thank you for submitting your article "Munc13 supports vesicle fusogenicity after disrupting active zone scaffolds and synaptic vesicle docking" for consideration by *eLife*. Your article has been reviewed by 3 peer reviewers, and the evaluation has been overseen by a Reviewing Editor and Lu Chen as the Senior Editor. The reviewers have opted to remain anonymous.

Essential revisions

1. It is unclear why Cre mediated removal of the presynaptic scaffold proteins was initiated at DIV 5, at a time point when synapse formation is already well underway. This likely contributed to the hypomorphic phenotype. Please comment.

2. The authors conclude that postsynaptic response is intact. On the other hand, they find kinetic changes in EPSC and IPSC (Figure 1 and Figure 4) that deserve to be discussed. In addition, they do not show any kinetic analysis of mEPSC and mIPSCs; this type of analysis is straightforward to be obtained and it might be useful to assess in part if there are or not changes at the postsynaptic level.

3. The major conclusion of the study is that the remaining release after ablation of RIM+ELKS (R+E) is mainly Munc13 dependent. First, the relative changes in mean EPSC/IPSC amplitude or sucrose-dependent RRP size between R+E and RIM+ELKS+Munc13 (R+E+M) KO cultures are comparably small, at most around 15% (Figures 4, 5). This is expected given the strong reduction in Munc13-1 levels in R+E KO (Figure 1; Wang et al., 2016). The relative differences between the experimental groups are smaller than the differences between different control data sets. Moreover, previous data sets obtained from R+E KO cultures under similar conditions by the same group suggest a similar relative decrease in E/IPSC amplitude or RRP size in R+E KO (Wang et al., 2016; Tan et al., 2022) compared to R+E+M KOs (both by ~90%). Given the parameter distributions (e.g., control EPSC amplitudes range between 0.5 and >1.5 nA and display clusters that likely reflect different cultures, Figure 4H), how confident can the authors robustly resolve average relative changes by ~10%? To evaluate whether the remaining release in R+E KOs is indeed Munc13-dependent, it would be helpful to report effect sizes and to provide a post-hoc power calculation. Irrespectively of the outcome of a power analysis, it may be advisable to increase sample sizes for some of the major experiments.

4. Although EPSC/IPSC amplitude, RRP size, and mEPSC/mIPSC frequency are strongly reduced in R+E+M hextuple KO cultures, these cultures still display significant evoked and spontaneous synaptic transmission. For instance, ~20% of the IPSC RRP (Figure 5H) or ~40% of mEPSCs (Figure 4B) remain in R+E+M KO neurons. The authors attribute this to incomplete loss of Munc13 or Munc13-independent release (Discussion). In line with the first hypothesis and previous reports, their confocal data of Munc13-1,-2,-3 KO cultures indicate a significant anti-Munc13-1 signal (~25% of control; Figure 1 —figure supplement 2B). Thus, it remains unclear whether the remaining release in R+E KO neurons is Munc13 dependent. An alternative hypothesis is that a significant fraction, if not the majority, of the remaining release is Munc13 independent. This would be equally, if not more interesting. Thus, unless the authors directly demonstrate that the release remaining in R+E KO neurons indeed requires Munc13, all respective statements in the manuscript should be revised accordingly (e.g., "Our data reveal that the transmitter release that remains after active zone disruption upon RIM and ELKS deletion depends on Munc13., l. 324)". Indeed, an alternative interpretation of the current data could be quite similar to that of Wang et al. 2016: Fusion competent vesicles persist upon ablation of RIM-1/2, ELKS^-1^/2, and Munc13-1/2.

5. The rise time of NMDAR EPSCs was previously shown to be strongly attenuated in R+E KO cultures (Wang et al., 2016). How does the EPSC rise time in R+E+M KO cultures relate to the one in R+E KO neurons?

6. The paired-pulse ratio data does not suggest changes in release probability (pr) between R+E and R+E+M conditions (Figure 6). Could the authors provide an additional, independent pr (and RRP?) estimate, e.g., based on (IPSC?) trains? Moreover, it would be interesting to plot the relative decrease in mini frequency for R+E KO and R+E+M KO and discuss how a potential change would relate to pr.

7. Regarding the quantification of STED fluorescence intensity data: Can the authors exclude that crosstalk between both channels causes the remaining Munc13-1 fluorescence in R+E+M cultures (Figure 1J, 2I)? How do the changes in fluorescence intensity at STED resolution compare to the corresponding changes at confocal resolution? What is the justification for restricting the analysis to side-view synapses? Finally, it would be helpful to plot relative changes in fluorescence intensity for R+E and R+E+M, similar to Figure 1 —figure supplement 1F.

[Editors' note: further revisions were suggested prior to acceptance, as described below.]

Thank you for resubmitting your work entitled "Munc13 supports vesicle fusogenicity after disrupting active zone scaffolds and synaptic vesicle docking" for further consideration by *eLife*. Your revised article has been evaluated by Lu Chen (Senior Editor) and a Reviewing Editor.

The manuscript has been improved but there are some remaining issues that need to be addressed, as outlined below:

*Reviewer #2 (Recommendations for the authors):*

The authors have significantly improved the manuscript with new experimental results and further substantial discussion of key aspects of the study. They have satisfactorily attended to my initial concerns.

*Reviewer #3 (Recommendations for the authors):*

In general, most points have been addressed by new experiments and text revisions. However, some of the concerns were apparently misunderstood by the authors. We would appreciate if the following remaining points could be clarified:

1. Regarding the major finding and effect sizes: Release is strongly compromised in both, RE KO (IPSC amplitude by ~82% of control) and REM KO cultures (~89% of control). EPSCs or IPSC amplitudes recorded in RE cultures were reduced by 90% and 81% (Wang et al., 2016), or by ~90% and ~85% (Tan et al., 2022) compared to time-matched controls in previous papers by the same group. Hence, the question arises whether the relative release defect of REM compared to RE cultures observed in the present study would persist in subsequent data sets.

Furthermore, the authors base their interpretation and effect size estimation on data that are normalized to the mean of the respective control group. Crucially, although the relative difference seems very high (~60%, comparing 11% with 18%), it strongly depends on the magnitude of the release defect. Conversely, the authors could have compared the relative reduction by 80% vs. 90% instead of the remaining fraction, which would give a relative difference of only ~13%. Thus, power calculations based on the relative effects of data normalized to the control mean are not informative in this context. Moreover, basing the sample sizes of the present study on previous studies with effect sizes of 80-90% appears inappropriate as well. To assess if release is indeed more strongly reduced in REM cultures, the authors could perform a two-way ANOVA and test for an interaction effect after not only normalizing the RE and REM datasets to the respective control means but also the control groups to their respective mean values. This analysis would reveal if there were indeed a significant difference between RE and REM. Otherwise, new data has to be collected to support this central finding of the paper.

2. Based on their data, the authors put forward a model in which the remaining release in RE KOs "depends on" Munc13. As most readers will unlikely know/look up the definition of "depend on", we strongly suggest using the term "partially depends on" (or the likes) throughout the text to clearly emphasize that they cannot differentiate between Munc13-dependent and independent mechanisms.

3. In their response, the authors claim that the relationship between confocal and STED fluorescence intensity measurements has been characterized extensively in previous papers (for example in Figures 1 and S1 of (Wong et al., 2018) and Figure S2 of (Tan et al., 2022)). However, data showing a clear correlation between confocal and STED fluorescence intensities were not presented in these references. Although we do not doubt the major conclusions based on the STED data, the authors should either cite another paper or provide a corresponding correlation in the present manuscript.

---

## [Author Response]

Essential revisions1. It is unclear why Cre mediated removal of the presynaptic scaffold proteins was initiated at DIV 5, at a time point when synapse formation is already well underway. This likely contributed to the hypomorphic phenotype. Please comment.

We thank the reviewers and editors for giving us the opportunity to comment, and we have performed a new experiment to address this important point. As a matter of background, in the culture system we use here, functional synapses are not detected at this time point. Spontaneous release is typically detectable only at DIV8-DIV9 (Mozhayeva et al., 2002), and evoked release builds up after DIV10. We have established before that with DIV5 cre infection to remove Ca_V_2 channels (Held et al., 2020), for example, release never builds up to robust levels (see Figure S4A and S4B, in (Held et al., 2020)). Thus, we think that DIV5 infection is appropriate in general. This point has been added on lines 119-120 of the manuscript.

In a previous study, we have shown that even when the Munc13-1 allele used here is knocked out throughout development by constitutive deletion in the germline, there continues to be some release in cultured autaptic neurons (see Figure S3 in (Banerjee et al., 2022)). We now include data that this is very similar in the Munc13 conditional knockout mice in the cultured neurons with DIV5 cre lentivirus infection (cKO^M^ neurons, new Figure 1 —figure supplement 2). We think that together, these observations make it highly likely that the persistence of some release in cKO^R+E+M^ neurons is not due to DIV5 cre infection, but to the allele that we use. This allele is described in detail in (Banerjee et al., 2022) and discussed in the present manuscript on lines 161-164.

We felt that the most important point was to establish that the persistence of synapse formation in the hextuple knockout neurons was not due to late infection. Therefore, we pursued an additional experiment in which Cre expression was initiated by lentiviral infection at DIV2 (Figure 2 —figure supplement 2). The density of synapses identified by Synaptophysin and PSD-95 was unaffected, indicating the synapse formation proceeds when RIM, ELKS and Munc13 are deleted with early cre infection.

2. The authors conclude that postsynaptic response is intact. On the other hand, they find kinetic changes in EPSC and IPSC (Figure 1 and Figure 4) that deserve to be discussed. In addition, they do not show any kinetic analysis of mEPSC and mIPSCs; this type of analysis is straightforward to be obtained and it might be useful to assess in part if there are or not changes at the postsynaptic level.

We thank the reviewers for bringing this up and have added analyses of kinetics of mEPSCs and mIPSCs (Figure 4 —figure supplement 1). There was a small increase of mEPSC rise times, similar to what we showed before in cKO^R+E^ neurons (Tan et al., 2022), while mEPSC decay times and mIPSC kinetics were unchanged. Overall, we think that together with the unchanged mPSC amplitudes, this argues that receptors are fundamentally there and detect neurotransmitter. It is currently not possible to fully exclude any postsynaptic effects, but we think in general they cannot be large effects dominating the phenotypes. We added this point on lines 223-226 and used wording that does not exclude that small effects may be present.

Kinetic changes in evoked IPSCs are indeed present in cKO^R+E^ synapses (as shown before (Wang et al., 2016)) and cKO^R+E+M^ synapses, and are presented in Figures 1 and 4. We think that they may reflect asynchrony of release, but drawing strong conclusions is not possible in the studied synapses, and we hope that the reviewers and editors agree that presenting the specific data point of increased IPSC rise times in a descriptive way is appropriate.

3. The major conclusion of the study is that the remaining release after ablation of RIM+ELKS (R+E) is mainly Munc13 dependent. First, the relative changes in mean EPSC/IPSC amplitude or sucrose-dependent RRP size between R+E and RIM+ELKS+Munc13 (R+E+M) KO cultures are comparably small, at most around 15% (Figures 4, 5). This is expected given the strong reduction in Munc13-1 levels in R+E KO (Figure 1; Wang et al., 2016). The relative differences between the experimental groups are smaller than the differences between different control data sets. Moreover, previous data sets obtained from R+E KO cultures under similar conditions by the same group suggest a similar relative decrease in E/IPSC amplitude or RRP size in R+E KO (Wang et al., 2016; Tan et al., 2022) compared to R+E+M KOs (both by ~90%). Given the parameter distributions (e.g., control EPSC amplitudes range between 0.5 and >1.5 nA and display clusters that likely reflect different cultures, Figure 4H), how confident can the authors robustly resolve average relative changes by ~10%? To evaluate whether the remaining release in R+E KOs is indeed Munc13-dependent, it would be helpful to report effect sizes and to provide a post-hoc power calculation. Irrespectively of the outcome of a power analysis, it may be advisable to increase sample sizes for some of the major experiments.

We thank the reviewers and editors for raising this point and for allowing us to comment on it. We have performed power analyses as requested and describe it below. We think that there are two important components to the point that is raised and we have made a concerted effort in the revised manuscript to be exceedingly clear on them.

Comparisons of controls across experiments. We think that these comparisons are complicated by a number of factors. The experiments are often months or years apart. There are genetic background differences between the mouse lines. In the specific case of the work presented here, there are genotype differences between the controls (the control^R+E+M^ neurons are Munc13-2 constitutive knockout neurons). Therefore, appropriate controls have to be performed for each experiment, which is now clearly described in the manuscript on lines 120-122, 147-152, 182-186. In the case of the presented experiments, this means that recordings are always done interleaved on the same day and with neurons that are identical in genotype except for the application of cre-expressing virus vs. Δcre-virus (a recombination-deficient version of cre). Hence, any cross-comparison of effects requires normalization to these controls, rather than direct comparison of absolute values, and controls should not be directly compared.

Effect magnitudes of control-normalized data. The only way to cross-compare is to normalize the mutant data to their controls that are matched in terms of genotype and interleafed on the day of recording. By doing so, in Figures 4 and 5, we compare means with 40-70% differences and assess significance based on this comparison. In these figures, the data are shown on a scale to 1.0 (100%) to reveal clearly that the data are normalized, but comparisons and statistics are done on effects sized >40%, not on 10-15% effect sizes. We are confident about these effects based on these numbers, and also based on the observation that they are present in EPSCs, IPSCs, and EPSC-RRP and IPSC-RRP measurements, which cross-validates the observations to some extent.

As instructed, we have also done post-hoc power analyses on these effects. Based on this analysis, the confidence to conclude that we detected an effect correctly in Figures 4L, 4M, 5I and 5J is 90.1%, 86.4%, 99.7% and 65.3% respectively. A different way to use post-hoc power analyses is to ask how many n’s we need minimally to detect the observed effect sizes with 90% confidence given the control data and its variability. We calculated this as well and found that we would need 16 cells in Figure 4L (actual experiment: 19 and 20 cells), 13 cells in Figure 4M (actual: 18 and 31 cells), 7 cells in Figure 5I (actual: 17 and 23 cells), and 9 cells in Figure 5J (actual: 18 and 21 cells). Altogether, these post-hoc calculations support that the number of observations we used in these experiments are sufficient to detect the observed effects with reasonable confidence.

Finally, we think that it is inappropriate to add n after an experiment is completed as we set n before we started the experiment. The only possible approach would be to repeat the entire experiment from the beginning independently with a larger n, which is not feasible for a revision (and we think not necessary given the points above). In summary, we feel that the effects are robust and established, and we hope that the reviewers and editors agree.

4. Although EPSC/IPSC amplitude, RRP size, and mEPSC/mIPSC frequency are strongly reduced in R+E+M hextuple KO cultures, these cultures still display significant evoked and spontaneous synaptic transmission. For instance, ~20% of the IPSC RRP (Figure 5H) or ~40% of mEPSCs (Figure 4B) remain in R+E+M KO neurons. The authors attribute this to incomplete loss of Munc13 or Munc13-independent release (Discussion). In line with the first hypothesis and previous reports, their confocal data of Munc13-1,-2,-3 KO cultures indicate a significant anti-Munc13-1 signal (~25% of control; Figure 1 —figure supplement 2B). Thus, it remains unclear whether the remaining release in R+E KO neurons is Munc13 dependent. An alternative hypothesis is that a significant fraction, if not the majority, of the remaining release is Munc13 independent. This would be equally, if not more interesting. Thus, unless the authors directly demonstrate that the release remaining in R+E KO neurons indeed requires Munc13, all respective statements in the manuscript should be revised accordingly (e.g., "Our data reveal that the transmitter release that remains after active zone disruption upon RIM and ELKS deletion depends on Munc13., l. 324)". Indeed, an alternative interpretation of the current data could be quite similar to that of Wang et al. 2016: Fusion competent vesicles persist upon ablation of RIM-1/2, ELKS^-1^/2, and Munc13-1/2.

We agree with the overall assessment, and we have made a concerted effort in the manuscript to be clear in phrasing. In respect to the cited statement above: we use the word “depends on” to express that release is further impaired in the RIM+ELKS+Munc13 knockouts, and we do not understand it equal to “is required for”. We think that this use is correct. For example, it is difficult to disagree with the following statement that we provide to illustrate our use of “to depend on”: “While our everyday lives depend a lot on email by now, email is not required for life”. Indeed, the Merriam-Webster definition of “depend on” is “to need (someone or something) for support”. We think that in that sense, the use of “depends on” is appropriate; our data reveal that the remaining release in cKO^R+E^ neurons depends on Munc13 because it is reduced by ~50% in cKO^R+E+M^ neurons. We have scrutinized the revised manuscript for the correct wording around this topic, and have changed “depends on” to “depends on at least partially” (or similar) in many cases, for example on lines 242, 261, 285, 315 and 349. The wording we use in captions is: “Munc13 contributes to a remaining functional RRP after active zone disruption” (results); “The remaining functional RRP in RIM+ELKS-deficient synapses depends on Munc13.” (Figure 5 caption).

The key question is of course why there is release left in cKO^R+E+M^. We think that there are two alternative, fundamentally different explanations for this point:

Incomplete loss of Munc13-1. A previous study has established in cultured autaptic neurons after germline deletion of Munc13-1 that excitatory synaptic transmission was impaired but not abolished in the allele we used here (Banerjee et al., 2022), different from previous Munc13-1 null alleles (Augustin et al., 1999). In a new experiment (Figure 1 —figure supplement 2), we confirm that this is also the case in dissociated mixed cultures and with application of cre-lentiviruses instead of germline recombination. The previous study (Banerjee et al., 2022) has described that a very small amount of spliced-over Munc13-1 remains after cre-recombination (estimated to be <5%). In our study, it was required to use the conditional allele because a hextuple knockout experiment would be excruciatingly complicated to do with a lethal allele. This is described on lines 161-164 of the manuscript.

Munc13-1 independent release. The alternative possibility is that some release remains because it is fully Munc13-independent. While this is an exciting possibility, it is impossible to make this conclusion in our experiments because there is a small amount of Munc13-1 left (point above) (Banerjee et al., 2022). We have included it as a possibility on lines 240-241 and 317-320 of the revised manuscript.

5. The rise time of NMDAR EPSCs was previously shown to be strongly attenuated in R+E KO cultures (Wang et al., 2016). How does the EPSC rise time in R+E+M KO cultures relate to the one in R+E KO neurons?

We thank the reviewers for pointing this out. The 20-80% rise time of NMDAR-EPSCs is significantly increased in cKO^R+E+M^ synapses compared with that in control^R+E+M^ synapses (control^R+E+M^: 7.6 ± 0.4 ms, cKO^R+E+M^: 9.1 ± 0.5 ms, p = 0.037, 20 cells/3 cultures each). This effect is qualitatively similar to cKO^R+E^ synapses. The effect magnitude is somewhat smaller here than in Ref. (Wang et al., 2016). Possible explanations are: (i) Munc13-2 is knocked out in control^R+E+M^ synapses but not in control^R+E^ synapses; (ii) different extracellular calcium concentrations are used here (1.5 mM) vs. before (2 mM) (Wang et al., 2016); (iii) other technical factors, for example cre infection time points or other differences between the current experiments and the ones that were done ~8 years ago (exact composition of some of the culture media ingredients). Finally, we note that the kinetics of these NMDAR responses are very slow (~5-10 ms rise kinetics) compared to the sub-millisecond kinetics of the fusion reaction. Hence, it is unclear how much of the slowdown reflects changes in release kinetics. Because of this uncertainty, we would prefer to not include the NMDAR kinetic analyses in the paper although they are slowed down and consistent with previous results. We believe that no interpretation should be made. We also think that this is different for IPSC kinetics, which are included in the manuscript because they are much faster and hence likely provide insight into release kinetics. We hope that the reviewers agree given our explanations.

6. The paired-pulse ratio data does not suggest changes in release probability (pr) between R+E and R+E+M conditions (Figure 6). Could the authors provide an additional, independent pr (and RRP?) estimate, e.g., based on (IPSC?) trains? Moreover, it would be interesting to plot the relative decrease in mini frequency for R+E KO and R+E+M KO and discuss how a potential change would relate to pr.

We thank the reviewers for these suggestions and have performed new experiments for both points.

Comparison of mini frequencies. For comparison of spontaneous synaptic transmission in cKO^R+E+M^ synapses and cKO^R+E^ synapses, we performed new recordings of mEPSCs and mIPSCs in control^R+E^ and cKO^R+E^ synapses (Figure 6 —figure supplements 1A, 1B, 1E, 1F; note that (Wang et al., 2016) used a different extracellular calcium concentration). At the same time, we performed new recordings of mEPSCs and mIPSCs in control^R+E+M^ synapses and cKO^R+E+M^ synapses such that the recordings were done during the same time frame and not years apart (Figures 4A-4F). This analysis reveals that the mini frequencies in cKO^R+E+M^ synapses and cKO^R+E^ synapses are similar (Figure 6 —figure supplements 1D and 1H). We also have previous datasets that were done simultaneously ~2 years ago (Figure 4A), allowing for an independent comparison of mEPSC frequencies in cKO^R+E+M^ and cKO^R+E^ synapses (the cKO^R+E^ results were published in (Tan et al., 2022)). In this earlier dataset as well, mEPSC frequencies were similar in the two genotypes.

Comparison of responses to stimulus trains. We now provide data recorded during train stimulation, and we find that the reduced depression during these stimulus trains is very similar in cKO^R+E^ synapses and cKO^R+E+M^ synapses (Figure 6 —figure supplements 1I-1L).

We think that overall, these data support the PPR data in Figure 6 to establish that release probability is not further affected by additional knockout of Munc13.

One often performed analysis is an RRP-determination based on back-extrapolation of trains, a method that might be pursued based on the reviewer comment above. We performed the analyses (Author response image 1), and overall, the outcomes look reasonable and consistent with all of our data. We note however, that assumptions that these methods require are not met by our data. Most importantly, vesicular release probability p has to be high in order for this analysis to be valid (Kaeser and Regehr, 2017; Neher, 2015; Thanawala and Regehr, 2016). In our mutants, p is severely affected after multi-protein knockout, and it cannot be rendered high enough, even by increasing extracellular calcium (Wang et al., 2016). A second important point is that the analysis should only be done on a homogenous population of synapses, which is not the case in our cultured hippocampal neurons. Hence, although the overall outcomes look reasonable, we strongly feel that the analyses should not be added to the paper because key assumptions are not met. We hope that the reviewers agree that just showing the IPSCs during stimulus trains (Figure 6 —figure supplements 1I-1L), but not the back-extrapolation (Author response image 1), is the most appropriate way to present these data.

**Author response image 1. sa2fig1:** (a, b) Cumulative IPSC amplitude plots in stimulus trains (original data from Figure 6 —figure supplements 1I-1L, n’s as in those figures). (c, d) Back-extrapolation to time zero yields the IPSC-amplitudes at Y-intercept to estimate the recovery-corrected pool size for each cell, back-extrapolation was based on the last ten (4150) responses. (e) Comparison of IPSC at Y-intercept normalized to their own controls in cKO^R+E^ (absolute data from c) and cKO^R+E+M^ (from d) neurons.

7. Regarding the quantification of STED fluorescence intensity data: Can the authors exclude that crosstalk between both channels causes the remaining Munc13-1 fluorescence in R+E+M cultures (Figure 1J, 2I)? How do the changes in fluorescence intensity at STED resolution compare to the corresponding changes at confocal resolution? What is the justification for restricting the analysis to side-view synapses? Finally, it would be helpful to plot relative changes in fluorescence intensity for R+E and R+E+M, similar to Figure 1 —figure supplement 1F.

We have performed two new experiments as outlined below. As a matter of background: we have used the method of side-view analyses extensively in the past five years and have published it in many papers (Emperador-Melero et al., 2021a, 2021b; Held et al., 2020; de Jong et al., 2018; Nyitrai et al., 2020; Tan et al., 2022; Wong et al., 2018). We also provide revised methods in the manuscript that refer to these papers and describe the methods in great detail. We note that we have also previously provided detailed supplemental figures on the justification and process of side-view synapse selection (Figure S3 in (Held et al., 2020), Figure S2 in (Nyitrai et al., 2020), Figure S6 in (Emperador-Melero et al., 2021b) and Figure S1 in (Tan et al., 2022)) and on the comparison between confocal and STED microscopy in this specific system (Figure S1 in (Wong et al., 2018), Figure S2 in (Tan et al., 2022)).

Crosstalk between channels. Based on our extensive experience with the use of this method for the characterization of knockouts, we are confident that there is no detectable crosstalk. To exclude the crosstalk for our specific conditions, we performed a new experiment in which we included a negative control without primary antibodies for the protein of interest, Munc13-1, with the same combination of primaries (Author response image 2). In the “no primary antibody” condition, no detectable signal in the relevant STED channel was detected, indicating that crosstalk is not the reason for background in our experimental setup.

**Author response image 2. sa2fig2:** (a-c) Sample STED images (a) and quantification (b, c) of side-view synapses stained for Munc13-1 (imaged in STED), PSD-95 (imaged in STED), and Synaptophysin (imaged in confocal). Munc13-1 primary antibody was not added during staining process as negative control (“no primary antibody”). Peak position and levels were analyzed in line profiles (600 nm x 200 nm) positioned perpendicular to the center of elongated PSD-95 structure and aligned to the PSD-95 peak, 20 synapses/1 cultures each.

Justification for side-view synapse selection. Side-view synapse selection is necessary because the analysis of active zone localization depends on assessing distances between the PSD and the active zone. This distance cannot be measured in synapses in other orientations, for example top-view, for two reasons. First, z-resolution in STED is worse than x-y resolution. Second, and critically, STED microscopy bleaches significantly in the area of signal acquisition. Hence, we only take single sections and do not reconstruct individual synapses in 3D. In a top-view synapse, the active zone and the PSD are overlayed in a single section and it is not possible to measure their distance. Hence, side-view selection is required. Because it is an experimenter-based selection process, this is done only based on the Synaptophysin and PSD-95 signals; at the time of side-view selection the experimenter is blind to the protein of interest. This is described in detail on lines 688-699.

Relationship of fluorescence intensities in confocal and STED imaging. We have characterized this in previous papers extensively, for example in Figures 1 and S1 of (Wong et al., 2018) and Figure S2 of (Tan et al., 2022). Hence, if there are changes in overall levels of a protein in a synapse, STED and confocal intensities typically correlate well. To directly compare Munc13-1 confocal (Figure 2 —figure supplements 1C and 1D) and STED levels (Figures 2G-2K) in cKO^R+E+M^ synapses, STED and confocal images stained for Synaptophysin, PSD-95 and Munc13-1 in control^R+E+M^ and cKO^R+E+M^ neurons were taken at the same time in new experiments. The Munc13-1 signals were removed efficiently at the active zone area in STED (Figures 2G-2K) and from synapses in confocal (Figure 2 —figure supplements 1C and 1D), and the reductions correlated well with one another.

Relative changes in fluorescence intensity for cKO^R+E^ and cKO^R+E+M^ synapses. We now provide this comparison in Figure 2 —figure supplement 1B. These analyses show that Munc13-1 levels in cKO^R+E^ synapses were higher than that in cKO^R+E+M^ synapses with a broad shoulder towards the inside of the nerve terminal, similar to what we observed in confocal microscopy (Figure 2 —figure supplement 1E).

We hope that these points, together with our previous extensive work on establishing this STED microscopy workflow, sufficiently answer the reviewers’ comments.

[Editors' note: further revisions were suggested prior to acceptance, as described below.]

The manuscript has been improved but there are some remaining issues that need to be addressed, as outlined below:Reviewer #3 (Recommendations for the authors):In general, most points have been addressed by new experiments and text revisions. However, some of the concerns were apparently misunderstood by the authors. We would appreciate if the following remaining points could be clarified:

We thank the reviewer for giving us the opportunity to clarify these remaining points.

1. Regarding the major finding and effect sizes: Release is strongly compromised in both, RE KO (IPSC amplitude by ~82% of control) and REM KO cultures (~89% of control). EPSCs or IPSC amplitudes recorded in RE cultures were reduced by 90% and 81% (Wang et al., 2016), or by ~90% and ~85% (Tan et al., 2022) compared to time-matched controls in previous papers by the same group. Hence, the question arises whether the relative release defect of REM compared to RE cultures observed in the present study would persist in subsequent data sets.Furthermore, the authors base their interpretation and effect size estimation on data that are normalized to the mean of the respective control group. Crucially, although the relative difference seems very high (~60%, comparing 11% with 18%), it strongly depends on the magnitude of the release defect. Conversely, the authors could have compared the relative reduction by 80% vs. 90% instead of the remaining fraction, which would give a relative difference of only ~13%. Thus, power calculations based on the relative effects of data normalized to the control mean are not informative in this context. Moreover, basing the sample sizes of the present study on previous studies with effect sizes of 80-90% appears inappropriate as well. To assess if release is indeed more strongly reduced in REM cultures, the authors could perform a two-way ANOVA and test for an interaction effect after not only normalizing the RE and REM datasets to the respective control means but also the control groups to their respective mean values. This analysis would reveal if there were indeed a significant difference between RE and REM. Otherwise, new data has to be collected to support this central finding of the paper.

After discussion of this point with the reviewing and senior editors for obtaining guidance, we have consulted with three biostatistics experts. As outlined below, we have added new statistical analyses. We thank the reviewer for insisting on this point, as we think that the newly added analyses, described below (and presented in the paper in Figures 4N, 4O, 5K, 5L, 6K, 6L, Figure 4 —figure supplement 2, Materials and methods) are a significant addition to this paper and hopefully for the field for better accounting of the data structure that is inherent to the types of analyses we perform.

Consultation. We consulted with the following experts. (1) Dr. Clement Ma is a Biostatistician and Assistant Professor at the Dalla Lana School of Public Health University of Toronto, he is also Co-Director of the Biostatistics Core Services at the Centre for Addiction and Mental Health. (2) The Harvard Biostatistics Student Consulting Center is a service for Harvard Researchers to consult about statistical testing. (3) Dr. Richard Born is a faculty member in the Neurobiology Department at Harvard and an expert in the quantitative analyses of neuroscience data. Dr. Born teaches all Harvard neuroscience graduate students in statistics, and this is a program with a heavy emphasis on quantitative methods. Graduates regularly go on to have careers in highly quantitative fields such as artificial intelligence research and computational neuroscience. The overall outcomes of these consultations were as follows:

a) The use of conventional statistical testing after control-normalization is appropriate as long as we only reach conclusions for the comparison of the two mutant groups, but not for control groups. We think that this is exactly how we used these comparisons. The controls are different (genetic background, constitutive knockout of Munc13-2) and should not be directly compared unless an experiment is designed to do so.

b) One general limitation in our analyses is that we compare means of individual measurements, and the statistical methods assume that the measurements are independent. However, this is not necessarily the case ^1^, We record multiple sweeps from each cell and multiple cells from each *batch* of culture, and we then average per cell. To account for this nested structure, we adapted an analysis method that does not suffer from this limitation and offers a fully independent test of the hypothesis that additional knockout of Munc13 impairs release over the deficit observed in RIM+ELKS knockout. The null hypothesis is that no further deficit is observed. We used a hierarchical bootstrap to test the probability of the null hypothesis. This method is similar to other hierarchical statistical approaches, such as repeated measures ANOVA and linear mixed effects models, however it has the added advantage of not making any distributional assumptions about the underlying data and it allows for unequal variance among the groups. In addition, it is much more intuitive and straightforward to implement even when there are multiple nested levels to the data – see ref. ^1^ for details and a comparison with other methods. These new analyses directly address the overarching concern that the detected reduction through the use of normalization and conventional statistics across four data sets may be explained by factors other than genotype.

Description of the hierarchical bootstrap. Two gene families involved in synaptic release (RIM and ELKS) are removed, which reduces synaptic transmission by about 80%. Now, it is addressed whether knockout of a third gene family, Munc13, further reduces synaptic transmission over this ~80% reduction. However, since the conditional RIM+ELKS knockout is in a different mouse *strain* from the conditional RIM+ELKS+Munc13 knockout, the measured parameters of synaptic transmission in each conditional knockout *strain* must be compared to their own controls. In each *strain*, the controls and test *conditions* are genetically identical except for the absence/presence of Cre recombinase.

We define: *strain* 1: R+E (short for RIM+ELKS)

*strain* 2: R+E+M (short for RIM+ELKS+Munc13)

To perform an experiment in a given *strain*, the hippocampi of several newborn mice are dissected out, and the *cells* are dissociated and pooled together in one primary culture. This culture is then divided into two pools subjected to different *conditions*: one is treated with a lentivirus containing Cre (= cKO); the other is treated with a lentivirus containing a catalytically inactive version of Cre (= control). From each culture, multiple *cells* are tested. Each *cell* is patch-clamped and synaptic transmission is tested by measuring the size of the postsynaptic current (EPSC or IPSC) evoked by an action potential or by the application of hypertonic sucrose. This measurement (*sweep*, technical replicate) is repeated 5-6 times for each *cell* for action potential-evoked release, and once per *cell* for sucrose-evoked release. The culture procedure is typically repeated in at least three *batches* of culture.

We define:

*condition* 1: cKO

*condition* 2: control

Thus, each measurement is uniquely identified by five numbers, which are variables (columns) in the tabulated data:

*strain*, R+E and R+E+M

*condition*, cKO and control

*batch*, always 3 *batches* per experiment

*cell*, may vary for different *batches*, 6 to 12 *cells* per *batch*

*sweep*, 5 or 6 per *cell* (action potential-evoked) or 1 per *cell* (sucrose-evoked)

Four experimental *groups* are defined:

cKO^R+E^

control^R+E^

cKO^R+E+M^

control^R+E+M^

The scientific question is whether knocking out Munc13 in addition to RIM+ELKS (cKO^R+E+M^) causes a greater relative decrease (vs. control^R+E+M^) in synaptic transmission as compared to cKO^R+E^. So, we define our test statistic, T, as:

T = [mean(cKO^R+E^) / mean(control^R+E^)] / [mean(cKO^R+E+M^) / mean(control^R+E+M^)] (1)

The null value for this statistic is: T = 1, corresponding to no further reduction,

and the alternate hypothesis is: T > 1, corresponding to a reduction.

In the hierarchical bootstrap ^1^, we estimate the sampling distribution of T by resampling with replacement from the raw data, while preserving the hierarchical relationships created by the design of the experiment. For each bootstrap iteration, *group* identity (combination of *strain* and *condition*) is preserved. Resampling is done at three nested levels: *batch*, *cell* and *sweep*. In the framework of a linear mixed effects model, *group* (= *strain* x *condition*) would be a fixed effect and *batch*, *cell* and *sweep* would be random effects. We developed a MATLAB code to perform the hierarchical bootstrap (https://github.com/kaeserlab/Hierarchical_Bootstrap_Analysis_RB).

For each of the four experimental *groups*, we perform the following hierarchical resampling. Starting with cKO^R+E^, we first resample with replacement from the *batches* of *cells* in this experiment. In all cases, there were three *batches* per experiment, so we use the MATLAB command 'unidrnd(3,3,1)' to draw 3 random samples from the uniform discrete distribution from 1 to 3. We might, for example, draw B_2_, B_2_, B_1_ for this iteration (Figure 4—figure supplement 2c).

We then start with *batch* 2, determine the number of *cells* sampled in this *batch*, and randomly resample from the *cells*, replicating the same number of *cells* but containing a different combination from that of the actual experiment. If there were 7 *cells* tested in *batch* 2, we might draw the following bootstrap sample: C_1_, C_3_, C_7_, C_6_, C_7_, C_5_, C_6_, in which *cells* 6 and 7 are included twice and *cells* 2 4 not at all. Then we proceed through this list of *cells*, each time randomly resampling (always with replacement) from the set of technical replicates for that *cell* and appending these resampled measurements to the bootstrap sample for this *group*. After doing this for each of the 7 *cells*, we go back and repeat the entire process, first for *batch* 2 (again), but selecting a different sample of the 7 *cells*, and then for *batch* 1. At the end we have a bootstrap sample for this experimental *group* that is exactly the same size as our original data set, but it contains a different subset of measurements and, importantly, it preserves the nested, hierarchical structure of our experiment. The above process is then repeated for the remaining three *groups* (control^R+E^, cKO^R+E+M^, control^R+E+M^), then T* is calculated using the formula (1) (the '*' denotes a bootstrap replicate of our test statistic). This entire procedure is repeated 100,000 times, producing an estimate of the sampling distribution of T. Based on the distribution of the 100,000 T* values, we calculated 95% confidence intervals using the percentile method ^2^. To do this, we sorted the 100,000 values of T* in ascending order, then defined the lower bound as the 2,500^th^ value and the upper bound as the 97,500^th^ value. In addition, we calculated the probability of the null hypothesis, P_H0_, given our data as:

P_H0_ = #{*T** ≤ 1} / 100,000

Note that this is not a traditional p-value calculated with standard statistical tests, which is the probability of obtaining a result as extreme or more extreme assuming the null hypothesis to be true, but rather a measure of the probability of a given hypothesis being true given our data ^1^.

Outcome. Using this method, we found low probabilities for the null hypotheses to be true: 0.042 for action potential-evoked EPSCs (Figure 4N), 0.009 for action potential-evoked IPSCs (Figure 4O), < 0.001 for sucrose-evoked EPSCs (Figure 5K), and 0.084 for sucrose-evoked IPSCs (Figure 5L). This confirms our conclusions reached with conventional statistics that synaptic transmission and the pool of releasable vesicles are further reduced by additional removal of Munc13.

Additional assessment of data structure. It seems a priori plausible that the design of the experiments would impart structure to our data. That is, it is likely that repeated measurements (*sweeps*) from one *cell* are more similar to each other than to measurements from another *cell*. And we might also expect that the *cells* from one *batch* are more similar to each other than to those from other *batches*, despite our best efforts to standardize the treatment of different *batches*. But is this in fact the case? Might it still be legitimate to treat all of the measurements from a given experimental *group* as independent?

To address this question, we performed two additional analyses. First, within each experimental *group* (i.e. eliminating the fixed effects of *strain* and *condition*), we performed 2-way ANOVA with *batch* and *cell* as factors. An example distribution of data points for the NMDAR-EPSC is shown in Figure 4—figure supplement 2b. For the two experiments (action potential-evoked EPSC and IPSC) in which there were repeated measurements for single *cells*, we ran the ANOVA for each of the four *groups*, yielding 8 tests, and found that *cell* was a significant factor in all 8 of them (at p < 0.001) and *batch* was a significant factor in 7 of 8 (also at p < 0.001). Including all four experiments (the above two experiments plus the sucrose-evoked EPSCs and IPSCs), we found *batch* to be a significant factor less frequently, but there was still evidence for higher between-*batch* variance than within-*batch* variance in 50% of the 16 cases (p < 0.05). This is not surprising, because each *batch* represented a mixture of neurons from several animals and were thus likely to be more homogeneous. Nevertheless, the ANOVA provides strong evidence that it is right to consider structure in the data to begin with and hence strengthen the point that a method that accounts for such structure, like the hierarchical bootstrap, should be used for analyses.

The second analysis was to repeat the bootstrap in a non-hierarchical way and compare the distributions of T* and the resulting confidence intervals with those from the hierarchical bootstrap. To do this, we simply pooled the measurements from all of the *batches* and *cells* from a given experimental *group*, and resampled with replacement from this pool. This instantiates the assumption that all measurements were independent. As expected, these distributions were much narrower and more bell-shaped than the corresponding distributions from the hierarchical bootstrap (Author response image 3), with the resulting 95% confidence intervals being, on average, less than half the width of their hierarchical counterparts. Importantly, extreme values of T* were much less likely to be represented in the non-hierarchical distribution, and the significance level of the results was overestimated. While this analysis does not directly address the issue of structure in the way that the ANOVA does, it provides a vivid picture of the consequences of not analyzing the data in the appropriate way, and further justifies the use of hierarchical bootstrap as a conservative method to judge the effects assessed here.

**Author response image 3. sa2fig3:** Comparison of hierarchical vs. non-hierarchical bootstrap for the analyses of NMDAR-EPSCs.

Assessment of paired pulse ratios with hierarchical bootstrap. Using the same methodology, we also analyzed the paired pulse ratios presented in Figure 6. Using conventional statistics, these ratios were found to be similar between *groups* (cKO^R+E^ and cKO^R+E+M^). With the hierarchical bootstrap, we found the probability of the null hypothesis (T ≥ 1) given our data was 0.69 (NMDAR-EPSC paired pulse ratio, Figure 6K), and 0.61 (IPSC paired pulse ratio, Figure 6L). Thus, not only did we fail to reject the null hypothesis by conventional statistics, but we found strong evidence in support of it, confirming that vesicular release probability is not further affected by removal of Munc13 in cKO^R+E^ synapses.

Power calculations. The power calculations (previous response, page 4) were done based on all data including the control conditions. Even though the data were normalized for the calculation, the control values were included in the calculation and the outcomes are hence independent of whether they are based on normalized or absolute data, all groups were part of the calculation. They are accurate regardless of normalization. We apologize that we did not explain this clearly in the previous response. Finally, the hierarchical bootstrap analyses presented here further support the point that sampling was big enough, because our conclusions are supported with an independent analyses method that accounts for the other variables in the experiments.

Changes to the manuscript. The following changes to the manuscript were made presenting these new analyses:

– inclusion of outcomes of hierarchical bootstrap in Figures 4N, 4O, 5K, 5L, 6K, 6L

– discussion of outcomes in corresponding Results section (page 11, top)

– inclusion of a new figure to describe the analysis procedure, Figure 4 —figure supplement 2

– generation and uploading of a new MATLAB code to perform these analyses, https://github.com/kaeserlab/Hierarchical_Bootstrap_Analysis_RB

– detailed description of the methodology of the bootstrap analyses in the “Materials and methods section”, sub-section “Statistics”

We thank the reviewer for insisting on finding a better way to analyze the effects across mouse lines. We feel that we have adapted a method that independently confirms the major conclusions. We will definitely use this method in the future, and we hope that others will as well.

2. Based on their data, the authors put forward a model in which the remaining release in RE KOs "depends on" Munc13. As most readers will unlikely know/look up the definition of "depend on", we strongly suggest using the term "partially depends on" (or the likes) throughout the text to clearly emphasize that they cannot differentiate between Munc13-dependent and independent mechanisms.

We now use "partially depends on" (or similar) throughout the text.

3. In their response, the authors claim that the relationship between confocal and STED fluorescence intensity measurements has been characterized extensively in previous papers (for example in Figures 1 and S1 of (Wong et al., 2018) and Figure S2 of (Tan et al., 2022)). However, data showing a clear correlation between confocal and STED fluorescence intensities were not presented in these references. Although we do not doubt the major conclusions based on the STED data, the authors should either cite another paper or provide a corresponding correlation in the present manuscript.

We thank the reviewer for asserting that there is no doubt about the major conclusion. The best comparison for this point is provided in Figures S2A, S2G, and S2H of reference ^3^. This comparison is the most appropriate because RIM1 expression levels are directly controlled using a stronger (RIM1α_high_) or weaker (RIM1α_low_) promotor. We think that this experiment is the only experiment that directly controls expression levels for direct comparison of Author response image 4 to plot the STED data similar to the confocal data, and identical to the way we plot peak values in the current manuscript (for example in Figures 1J, 1K, 2E, 2F, 2J and 2K). This analysis reveals that levels assessed with STED microscopy are highly similar to levels assessed with confocal microscopy.

**Author response image 4. sa2fig4:** (a, b) Comparison of RIM1 levels between confocal and STED images. In a, RIM1 levels in indicated conditions are from confocal images (from Figure S2A of ^3^). In b, RIM1 levels in indicated conditions are from STED images (data from Figure S2H of ^3^).

We do not have a similarly valid comparison in the current manuscript with an experiment in which we actively titrate expression levels. In fact, there are no experiments with exogenous expression in the current manuscript at all. There are, however, experiments for which a strong correlation is expected. The best current experiment in this respect is the analysis of Munc13 antibody stainings in control^M^ and cKO^M^ neurons. These stainings are analyzed both with STED microscopy (Figure 1 —figure supplement 1F) and with confocal microscopy (Figure 1 —figure supplement 3E). The signal reductions are similar to one another with 28% remaining in confocal microscopy and 21% remaining in STED microscopy (we note that there is a small amount of Munc13-1 left in these mutants, ref. ^4^). We now state this in the supplemental figure legend of Figure 1 —figure supplement 3E.

References

1. Saravanan, V., Berman, G. J. and Sober, S. J. Application of the hierarchical bootstrap to multi-level data in neuroscience. *Neurons, Behav. data Anal. theory* 3, (2020).

2. Efron, B. and Tibshirani, R. J. *An Introduction to the Bootstrap*. (Chapman and Hall/CRC, 1994). doi:10.1201/9780429246593

3. Tan, C., Wang, S. S. H., de Nola, G. and Kaeser, P. S. Rebuilding essential active zone functions within a synapse. *Neuron* 110, 1498-1515.e8 (2022).

4. Banerjee, A. *et al.* Molecular and functional architecture of striatal dopamine release sites. *Neuron* 110, 248-265.e9 (2022).